

# Reconstructing partonic kinematics
# at colliders with machine learning

**David F. Rentería Estrada**[1*], **Roger J. Hernández-Pinto**[1],
**German F. R. Sborlini**[2,3] **and Pia Zurita**[4]

**1** Facultad de Ciencias Físico-Matemáticas, Universidad Autónoma de Sinaloa,
Ciudad Universitaria, CP 80000 Culiacán, Mexico
**2** Deutsches Elektronen-Synchrotron DESY, Platanenallee 6, 15738 Zeuthen, Germany
**3** Departamento de Física Fundamental e IUFFyM,
Universidad de Salamanca, E-37008 Salamanca, Spain
**4** Institut für Theoretische Physik, Universität Regensburg, 93040 Regensburg, Germany

★ davidrenteria.fcfm@uas.edu.mx

## Abstract

In the context of high-energy physics, a reliable description of the parton-level kinematics plays a crucial role for understanding the internal structure of hadrons and improving the precision of the calculations. In proton-proton collisions, this represents a challenging task since extracting such information from experimental data is not straightforward. With this in mind, we propose to tackle this problem by studying the production of one hadron and a direct photon in proton-proton collisions, including up to Next-to-Leading Order Quantum Chromodynamics and Leading-Order Quantum Electrodynamics corrections. Using Monte-Carlo integration, we simulate the collisions and analyze the events to determine the correlations among measurable and partonic quantities. Then, we use these results to feed three different Machine Learning algorithms that allow us to find the momentum fractions of the partons involved in the process, in terms of suitable combinations of the final state momenta. Our results are compatible with previous findings and suggest a powerful application of Machine-Learning to model high-energy collisions at the partonic-level with high-precision.

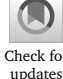

# 1 Introduction

Thanks to recent technological advances and increased computational power, Machine Learning (ML) has taken by storm our everyday life. Applications of ML cover fields as diverse as image and speech recognition, automatic language translation, product recommendation, stock market prediction and medical diagnosis, to mention some examples. High-energy physics has not remained indifferent to the opportunities offered by these techniques. In the last years several applications have been developed, particularly in regards to data analysis. Novel jet clustering algorithms that use improved classification to identify structures [1], reconstruction of the Monte-Carlo (MC) parton shower variables [2], and reconstruction of the kinematics [3,4] are just some of the explored uses. In particular, the high luminosity upgrade of the Large Hadron Collider (LHC), the upcoming Electron-Ion Collider (EIC), and the planned Future Circular Collider (FCC), International Linear Collider (ILC) and Compact Linear Collider (CLIC) are feeding the interest of the community in ML.[1] From a theoretical perspective there has been progress in the calculation of higher-order scattering amplitudes assisted by ML algorithms [5] and, in phenomenology, the determination of the partonic structure of hadrons greatly profited from powerful neural-network machinery (for instance, see the works of NNPDF collaboration in Refs. [6–13]). Furthermore, very recent implementations of ML algorithms in event generators and simulators for HEP were reported in Ref. [14].

 The successes of the perturbative expansion of Quantum Chromodynamics (QCD) to describe processes involving hadrons lies in the factorisation of the physical observables into hard (perturbative, process-dependent) and soft (non-perturbative, universal) terms [15]. The former describe the interaction between elementary particles while the latter encode all the information concerning non-perturbative physics, i.e., the description of the partons inside the hadrons before the interaction and their posterior hadronisation into detected particles. For these, only the scale evolution can be determined once they are known at some other scale,

---

[1]For more information concerning the LHC upgrade, we refer the reader to https://home.cern/science/accelerators/high-luminosity-lhc. Details about ML developments for the upcoming EIC were presented at the workshop *AI4EIC - Experimental Applications of Artificial Intelligence for the Electron Ion collider* (https://indico.bnl.gov/event/10699/), whilst the Inter-experiment Machine Learning (IML) group at LHC organizes a dedicated workshop series (https://indico.cern.ch/event/852553/).

and thus must be obtained from data through global fits.[2] The simplest description of a hadron is that of a collection of partons moving in the same direction. The probability of finding a particular parton $a$ in a hadron $H$ carrying a fraction $x$ of its momentum is given by the parton distribution function (PDF) $f_{H/a}(x, \mu)$, when the hadron is explored at scale $\mu$. After the hard interaction all outgoing coloured particles will hadronise; the probability of a parton $a$ to fragment into a hadron $H$ with a fraction $z$ of its original momentum is described by the fragmentation function (FF) $D_{a/H}(z, \mu)$. This collinear picture is the best explored and in this framework several sets of PDFs and FFs have been extracted using standard regression techniques (e.g. [17–21]), MC sampling (e.g. [22, 23]) and MC sampling with neural networks (e.g. [6]).

In order to perform a meaningful calculation, the hard cross-section must be convoluted with the PDFs and/or FFs, over the corresponding momentum fractions of the partons. In the inclusive deep inelastic scattering (DIS) process, where a lepton and a parton inside a hadron interact by exchanging momentum $Q^2 \geq 1 \text{ GeV}^2$, measuring the scattered lepton (and/or final hadrons) provides the full kinematics of the event. Unfortunately, in proton-proton ($p+p$) collisions the situation is not so simple. One has to estimate the momenta of the initial partons (that enter in the evaluation of the PDFs) using the measured momenta and scattering angles of the final state particles. Depending on the process and the characteristics of the detectors, it can become a complicated task. Despite its inherent complexity, it is of the utmost importance in some situations. For example in the case of asymmetric proton-nucleus ($p+A$) collisions, particles created in the the backward (nucleus-going) direction are linked to initial partons in the nucleus with low-$x$, and those in the forward (proton-going) direction are related to partons in the nucleus with large-$x$. Depending on its exact value, one could have an enhancement or a suppression of the nuclear PDF w.r.t. the free proton one. Knowing the region of the detector associated with the kinematics of interest for a given process is also relevant for the efficient design and construction of the detectors [24]. The proper mapping of the measured kinematics onto the partonic level is crucial for a correct evaluation of the cross-sections and interpretation of the perturbative calculations. This can be done analytically at leading order (LO) for processes involving few particles, but as one considers higher orders the emission of real particles makes it hard to fully determine the kinematics, and normally phenomenological approximations are used.

In the present work, we aim to use ML to determine the relation between the measurable four-momenta of the final particles and the parton-level kinematics. In particular, we focus on $p+p$ collisions with one photon plus one hadron in the final state, computed using QCD and Quantum Electrodynamics (QED) corrections. This process has already been identified as an interesting observable at the Relativistic Heavy-Ion Collider (RHIC) [25], and previous studies were performed with DIPHOX [26–28] (including up to NLO QCD corrections). Our goal is to obtain the functions that, depending on the four-momenta of the photon and hadron, give $x_i$ (the fraction of momentum of the proton $i$ carried by the parton coming from it, $i = 1, 2$) and $z$, the fraction of energy of the parton coming from the hard interaction that is taken by the hadron (in our analysis a pion).

This article is organised as follows. In Sec. 2 we describe the framework used to implement the MC simulation of hadron-photon production, with special emphasis on the isolation prescription (Sec. 2.1). Relevant phenomenological aspects of the process are discussed in Sec. 3. The distributions w.r.t. different variables are presented in Sec. 3.1, with the purpose of identifying the most probable configurations. We also explore the correlations between different measurable variables and the partonic momentum fractions in Sec. 3.2. In Sec. 4, we detail the implementation of reconstruction algorithms based on ML to approximate the

---

[2]Significant progress in the ab-initio calculation of parton densities is being carried out in the field of Lattice QCD [16].

partonic momentum fractions using only measurable quantities. Finally, we discuss the results and comment on potential future applications of our methodology in Sec. 5.

## 2  Computational setup

From the theoretical point of view, the calculation relies on the factorization theorem to separate the low-energy hadron dynamics (i.e. the non-perturbative component embodied into the PDFs and FFs) from the perturbative interactions of the fundamental particles. This approach is valid in the high-energy regime, under the assumption that the typical energy scale of the process is much larger than $\Lambda_{\text{QCD}} \approx 300\text{MeV}$. The process under consideration is described by

$$H_1(P_1) + H_2(P_2) \to h(P^h) + \gamma(P^\gamma),\tag{1}$$

and the differential cross-section is given by

$$
\begin{aligned}
d\sigma_{H_1 H_2 \to h\gamma} &= \sum_{a_1 a_2 a_3 a_4} \int dx_1\, dx_2\, dz_1\, dz_2\, f_{H_1/a_1}(x_1, \mu_I) f_{H_2/a_2}(x_2, \mu_I) D_{a_3/h}(z_1, \mu_F) \\
&\quad \times D_{a_4/\gamma}(z_2, \mu_F)\, d\hat{\sigma}_{a_1 a_2 \to a_3 a_4}(x_1 P_1, x_2 P_2, P^h/z_1, P^\gamma/z_2; \mu_I, \mu_F, \mu_R),
\end{aligned}\tag{2}
$$

where $\{a_i\}$ denote the possible flavours of the partons entering into the fundamental high-energy collision. $f_{H_i/a_j}(x, \mu_I)$ is the PDF of the parton at the initial state factorization scale $\mu_I$, and $D_{a_j/h}(z, \mu_F)$ is the FF of the parton at the final state factorization scale $\mu_F$. The partonic cross-section, $d\hat{\sigma}$, depends on the kinematics of the partons as well on the factorization and renormalization scales ($\mu_R$) and can be computed using perturbation theory. It is worth appreciating that we consider all the partons to be massless.

In Eq. (2) we consider the photon as a parton, i.e. $a_i \in \{q, g, \gamma\}$. Namely, we rely on the extended parton model to include mixed QCD-QED corrections in a consistent way [29–33]. However, we will assume that the fragmentation of a photon into any hadron is highly suppressed w.r.t. the same process initiated by a QCD parton. This implies that we neglect $D_{\gamma/h}$ and $a_3$ is always a QCD parton (quark or gluon). Also, since we are looking for a photon in the final state, we can write

$$D_{a_4/\gamma}(z_2, \mu_F) = \delta_{a_4,\gamma} \delta(z_2 - 1) + (1 - \delta_{a_4,\gamma}) \tilde{D}_{a_4/\gamma}(z_2, \mu_F),\tag{3}$$

which leads to

$$
\begin{aligned}
d\sigma_{H_1 H_2 \to h\gamma} &= \sum_{a_1 a_2 a_3} \int dx_1\, dx_2\, dz\, f_{H_1/a_1}(x_1, \mu_I) f_{H_2/a_2}(x_2, \mu_I) D_{a_3/h}(z, \mu_F) \\
&\quad \times d\hat{\sigma}_{a_1 a_2 \to a_3 \gamma}(x_1 P_1, x_2 P_2, P^h/z, P^\gamma; \mu_I, \mu_F, \mu_R) \\
&+ \sum_{a_1 a_2 a_3} \sum_{a_4 \in \text{QCD}} \int dx_1\, dx_2\, dz_1\, dz_2\, f_{H_1/a_1}(x_1, \mu_I) f_{H_2/a_2}(x_2, \mu_I) D_{a_3/h}(z_1, \mu_F) \\
&\quad \times \tilde{D}_{a_4/\gamma}(z_2, \mu_F)\, d\hat{\sigma}_{a_1 a_2 \to a_3 a_4}(x_1 P_1, x_2 P_2, P^h/z_1, P^\gamma/z_2; \mu_I, \mu_F, \mu_R),
\end{aligned}\tag{4}
$$

where $a_4$ is a QCD parton. By rewriting Eq. (2) in this way, it is possible to identify at least two mechanisms originating photons in the final state.[3] The first term describes the *direct* production of an observed photon in the partonic collision; in the second term the observed

---

[3]Another mechanism is related to the presence of fracture functions, $M_{a_3, a_4/h, \gamma}$, which do not completely separate the non-perturbative interactions in the final state. Since we are interested in the high-energy limit of this process, such contributions will be suppressed by the same reasons supporting the validity of the factorization theorem.

*resolved* photon is generated from a non-perturbative process initiated by the parton $a_4$. These contributions are not individually distinguishable; however the latter can be suppressed by applying adequate prescriptions. By realising that the resolved component appears in the context of hadronisation, the photon being produced together with a bunch of hadrons, one can exploit this signature to enhance the direct photon: it is the motivation for introducing *isolation prescriptions*. By selecting mainly those events that contain photons isolated from hadronic energy, the total cross-section can be approximated by

$$
\begin{aligned}
d\sigma_{H_1 H_2 \to h\gamma} \quad \approx \quad & \sum_{a_1 a_2 a_3} \int dx_1 \, dx_2 \, dz \, f_{H_1/a_1}(x_1, \mu_I) f_{H_2/a_2}(x_2, \mu_I) D_{a_3/h}(z, \mu_F) \\
& \times d\hat{\sigma}^{(ISO)}_{a_1 a_2 \to a_3 \gamma}(x_1 P_1, x_2 P_2, P^h/z, P^\gamma; \mu_I, \mu_F, \mu_R),
\end{aligned}
\tag{5}
$$

i.e. neglecting the resolved component and summing over all QCD-QED partons. The partonic cross-section $d\hat{\sigma}^{(ISO)}_{a_1 a_2 \to a_3 \gamma}$ incorporates the isolation prescription and is described in greater detail in Sec. 2.1.

We can now move to the discussion of how to include the QED corrections. The next-to-leading order (NLO) pure QCD corrections for this process were computed in Refs. [25, 34]. Since in this case we are dealing with mixed QCD-QED corrections, we have to consider the two couplings involved in the perturbative expansion. From the computational point of view, we can profit from the Abelianization techniques to directly obtain QED contributions from the QCD ones [31, 32, 35–37]. Given that the energy scale of the process is roughly $\mathcal{O}(10\,\text{GeV})$, we have $\alpha_S \approx 0.12$ and $\alpha \approx 1/129$. This means $\alpha \approx \alpha_S^2$, indicating that the LO QED corrections have the same weight as the NLO QCD ones. Therefore, the dominant contribution is given by the partonic channels $q\bar{q} \to g\gamma$ and $qg \to q\gamma$ at $\mathcal{O}(\alpha_S \alpha)$, i.e.

$$
d\hat{\sigma}^{ISO,(0)}_{a_1 a_2 \to a_3 \gamma} \quad = \quad \frac{\alpha_S}{2\pi} \frac{\alpha}{2\pi} \int dPS^{2\to 2} \frac{|\mathcal{M}^{(0)}|^2(x_1 P_1, x_2 P_2, P^h_3/z, P^\gamma)}{2\hat{s}} \mathcal{S}_2,
\tag{6}
$$

with $\mathcal{S}_2$ the measure function containing the definition of the kinematical selection cuts for the $2 \to 2$ sub-processes. We have then to include $\mathcal{O}(\alpha_S^2 \alpha)$ and $\mathcal{O}(\alpha_S \alpha^2)$ contributions, associated to the partonic channels

$$
q\bar{q} \to g\gamma g, \quad qg \to q\gamma g, \quad gg \to q\gamma\bar{q}, \quad q\bar{q} \to Q\gamma\bar{Q}, \quad qQ \to q\gamma Q,
\tag{7}
$$

and

$$
q\gamma \to q\gamma, \quad q\bar{q} \to \gamma\gamma,
\tag{8}
$$

respectively. $q$ and $Q$ are used to indicate two different quark flavours. In this way, the corrections to the partonic cross-section are given by [38]

$$
\begin{aligned}
d\hat{\sigma}^{ISO,(1)}_{a_1 a_2 \to a_3 \gamma} \quad = \quad & \frac{\alpha^2}{4\pi^2} \int dPS^{2\to 2} \frac{|\mathcal{M}^{(0)}_{QED}|^2(x_1 P_1, x_2 P_2, P^h/z, P^\gamma)}{2\hat{s}} \mathcal{S}_2 \\
& + \frac{\alpha_S^2}{4\pi^2} \frac{\alpha}{2\pi} \int dPS^{2\to 2} \frac{|\mathcal{M}^{(1)}|^2(x_1 P_1, x_2 P_2, P^h/z, P^\gamma)}{2\hat{s}} \mathcal{S}_2 \\
& + \frac{\alpha_S^2}{4\pi^2} \frac{\alpha}{2\pi} \sum_{a_r} \int dPS^{2\to 3} \frac{|\mathcal{M}^{(0)}|^2(x_1 P_1, x_2 P_2, P^h/z, P^\gamma, k_r)}{2\hat{s}} \mathcal{S}_3,
\end{aligned}
\tag{9}
$$

where $\hat{s}$ is the partonic center-of-mass energy and $r$ denotes the extra parton associated to the real radiation correction. $|\mathcal{M}^{(0)}|^2$ and $|\mathcal{M}^{(1)}|^2$ are the squared matrix-elements for the tree-level and one-loop corrections, respectively. In these expressions, $\mathcal{S}_3$ represents the measure

function that implements the experimental cuts and the isolation prescription for the $2 \to 3$ sub-processes.

Since we are dealing with higher-order corrections, singularities will appear in the calculation. The LO QED is given by a (finite) Born level process. However, the NLO QCD corrections involve both ultraviolet (UV) and infrared (IR) singularities that must be regularized and cancelled to get a physical result. The regularization was done using Dimensional Regularization (DREG) [39–42]. The virtual corrections were computed starting from one-loop QCD amplitude for the process $0 \to q\bar{q}g\gamma$, removing the UV poles through the renormalization in the $\overline{\text{MS}}$ scheme. In order to cancel the IR singularities, we relied on the subtraction formalism [43–47], splitting the real phase-space in regions containing only one kind of IR singularity. When combining the real and the virtual corrections, some of the IR divergences associated to final state radiation (FSR) cancel by virtue of the KLN theorem [48,49]. But to achieve a full cancellation, counter-terms were added to remove the remaining initial-state and final-state contributions absorbed into the PDFs and FFs, respectively. In this way, the master formula for the partonic cross-section at NLO QCD + LO QED accuracy is symbolically given by

$$
\begin{aligned}
d\hat{\sigma}^{\text{ISO},(1),\text{finite}}_{a_1 a_2 \to a_3 \gamma} &= d\hat{\sigma}^{\text{ISO},(1),\text{ren.}}_{a_1 a_2 \to a_3 \gamma} - \frac{C^{\text{UV}}_{a_1 a_2 \to a_3 \gamma}}{\epsilon} \times d\hat{\sigma}^{\text{ISO},(0)}_{a_1 a_2 \to a_3 \gamma} \\
&\quad - d\hat{\sigma}^{\text{ISO},\text{cnt},(I)}_{a_1 a_2 \to a_3 \gamma} - d\hat{\sigma}^{\text{ISO},\text{cnt},(F)}_{a_1 a_2 \to a_3 \gamma},
\end{aligned}
\tag{10}
$$

where $d\hat{\sigma}^{\text{ISO},\text{cnt},(I)}_{a_1 a_2 \to a_3 \gamma}$ and $d\hat{\sigma}^{\text{ISO},\text{cnt},(F)}_{a_1 a_2 \to a_3 \gamma}$ are the initial and final-state IR counter-terms, respectively. Here, $C^{\text{UV}}_{a_1 a_2 \to a_3 \gamma}$ is the renormalization counter-term for the partonic process $a_1 a_2 \to a_3 \gamma$ in the $\overline{\text{MS}}$ scheme.[4]

## 2.1 Isolation prescription and other assumptions

In order to suppress events with photons originated from the decay of hadrons, it is necessary to implement an isolation prescription. The idea behind most of the strategies available in the literature consists in quantifying the amount of hadronic energy surrounding a well-identified photon, and rejecting events with more hadronic energy than a certain threshold. Whilst most of the prescriptions work nicely at LO, not all of them are infrared safe. For instance, it is known that choosing a fixed cone eliminates events that play a crucial role in the cancellation of IR singularities. Thus, special care is needed in the implementation of these methods.[5]

In this work, we rely on the smooth cone prescription introduced in Ref. [54]. Its main advantage is that it suppresses the resolved component without preventing the emission of soft/collinear QCD radiation, which makes it IR-safe and fully suitable for higher-order calculations. In the first place, we fix a reference point in the rapidity-azimuthal plane $(\eta_0, \phi_0)$, and define the distance

$$
r(j) = \sqrt{(\eta_j - \eta_0)^2 + (\phi_j - \phi_0)^2},
\tag{11}
$$

with $(\eta_j, \phi_j)$ the angular coordinates of the parton $j$. Once we identify a photon in the detector, we trace a cone of radius $R$ around it and look for QCD partons inside. If no QCD radiation lays inside the cone, the photon is isolated. If not, we identify the QCD partons inside the cone, $\{a_j\}$, and measure their distance to the photon following Eq. (11). Then, for a fixed $r \leq R$, we calculate the sum of the hadronic transverse energy according to

$$
E_T(r) = \sum_{r_j \leq r} E_{T_j}.
\tag{12}
$$

---

[4]Explicit formulae for all the ingredients in this expression can be found in Refs. [45,50].

[5]An extensive study of different methods and their impact on the calculations is available in Refs. [51–53].

We want to restrict $E_T$ by imposing an upper bound, thus limiting the amount of hadronic energy surrounding the photon. In the fixed cone prescription, this limit is a constant. However, for the smooth prescription, we introduce an arbitrary smooth function $\xi(r)$ satisfying $\xi(r) \to 0$ for $r \to 0$, and require $E_T(r) < \xi(r)$ for every $r < r_0$. Only if this condition is fulfilled, the photon is isolated; otherwise, the event is rejected.

The experimental implementation of this criterion requires a very high angular resolution, something that is usually not achievable in practise. This is one of the reasons why most of the current experiments still rely (mainly) on the fixed cone prescription. Fortunately, the difference between both approaches can be neglected for several relevant observables [51, 52]. In any case, technological improvements in detector science will certainly reduce the experimental limitations in the near future.

Finally, let us mention one further detail about the implementation. We will neglect the partonic channel $q\bar{q} \to \gamma\gamma$ in Eq. (5), which would imply the introduction of the fragmentation $D_{\gamma/h}$. From the point of view of perturbation theory, this fragmentation can be interpreted as a collinear electromagnetic splitting $\gamma \to a + X$, with $a$ a QCD-parton that undergoes hadronization to generate the observed hadron $h$. Performing a naive counting, this contribution is $\mathcal{O}(\alpha^3)$ and turns out to be sub-leading w.r.t. the NLO QCD + LO QED terms studied in this work.[6]

## 3 Phenomenological results

Using the formalism explained in the previous Section, we calculated the unpolarized cross-section via a code that uses adaptive MC integration. In this program, the different contributions to $2 \to 2$ and $2 \to 3$ processes are computed independently, and kinematic cuts can be imposed. The events are randomly generated, using a different seed for each contribution: we collect all the events compatible with certain cuts to define the histograms.

We considered two different experimental scenarios. On one side, we simulated RHIC kinematics with centre-of-mass (c.m.) energy ($\sqrt{S_{CM}}$) 500 GeV and reproduced the cuts corresponding to the PHENIX detector, i.e.

$$|\eta^h| \leq 0.35, \quad |\eta^\gamma| \leq 0.35, \quad p_T^h \geq 2\,\text{GeV}, \quad 5\,\text{GeV} \leq p_T^\gamma \leq 15\,\text{GeV}, \tag{13}$$

with $\eta$ the rapidity of the particles measured in the hadronic c.m. frame. On top of that, we require $|\phi^h - \phi^\gamma| > 2$ to retain those events with the photon and hadron produced almost back-to-back. On the other side, we simulated LHC Run II kinematics with $\sqrt{S_{CM}} = 13$ TeV. Regarding the detector cuts, we kept the same restrictions for $\eta^h$ and $p_T^h$ given in Eq. (13). However, the intensive hadron activity and pile-up associated to the high-luminosity LHC Run II might contaminate the photon selection. Both ATLAS and CMS have performed dedicated analyses of the trigger efficiency for different photon energies [55–57], showing a better triggering and reconstruction efficiency for $E_T^h > 30$ GeV. For this reason, we set

$$|\eta^\gamma| \leq 2.5, \quad p_T^\gamma \geq 30\,\text{GeV}, \tag{14}$$

when simulating LHC Run II kinematics. The extended rapidity range w.r.t. PHENIX is due to the geometry of the detector, even if there is a gap for $1.36 < |\eta^\gamma| \leq 1.55$. For the sake of simplicity, we ignore this gap and assume that photons can be efficiently detected and reconstructed when they fulfil Eq. (14).

---

[6]This topic deserves attention, specially because non-perturbative contributions could enhance the production rate of hadrons from highly-energetic photons. Unfortunately, we were unable to find in the literature studies or a proper definition of $D_{\gamma/h}$ to be included within our simulations.

Regarding the non-perturbative ingredients of the calculation, we used the LHAPDF package [58, 59] to have a unified framework for the PDF implementation. We relied on the NNPDF4.0NLO [13] and NNPDF3.1luxQEDNLO [60–63] parton distributions for the pure QCD and mixed QCD-QED calculations, respectively. In both cases, we use the set 0, which corresponds to an average over the different replicas. For the fragmentation functions, we used the DSS2014 set at NLO accuracy [20, 64]. Since the pion is the lightest hadron and is produced more copiously, we restrict our attention to the case $h = \pi^+$. Also, we evolve the QCD and QED couplings using the one-loop RGE with the initial conditions $\alpha_S(m_Z) = 0.118$ and $\alpha(m_Z) = 1/128$.

Finally, we fixed the factorization and renormalization scales to be equal to the average transverse momenta of the hadron and the photon, i.e.

$$\mu_F = \mu_I = \mu_R = \frac{p_T^\pi + p_T^\gamma}{2}. \tag{15}$$

Regarding the implementation of the smooth isolation criteria, we used the function

$$\xi(r) = E_T^\gamma \left( \frac{1 - \cos(r)}{1 - \cos(r_0)} \right)^4, \tag{16}$$

where $E_T^\gamma$ is the transverse energy of the photon and $r_0 = 0.4$. As mentioned before, the only requirement for $\xi(r)$ is that $\xi(r) \to 0$ for $r \to 0$, and Eq. (16) fulfils this condition.

## 3.1 One-dimensional distributions

Since we are looking at the process $p + p \to \pi + \gamma + X$,

$$\mathcal{V}_{\text{Exp}} = \{p_T^\gamma, p_T^\pi, \eta^\gamma, \eta^\pi, \cos(\phi^\pi - \phi^\gamma)\}, \tag{17}$$

are the experimentally accessible variables measured in the c.m. system. Notice that we consider only the difference of the azimuthal angles, because the problem has rotational symmetry around the collision axis. Moreover, it turns out that $\cos(\phi^\pi - \phi^\gamma)$ is a variable often used by experimental collaborations [65].

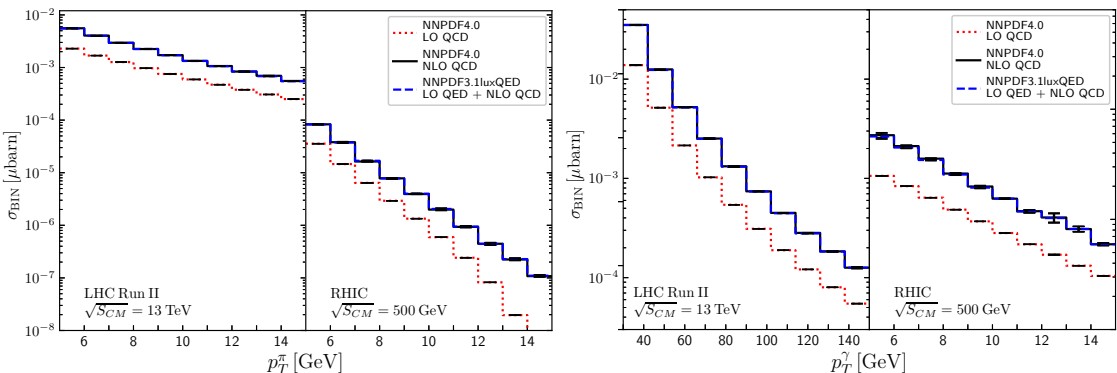

Figure 1: Unpolarized cross-section for the production of one photon plus one pion as a function of the transverse momentum of the pion (left) and the photon (right), respectively. We considered the selection cuts described in the previous section, for LHC Run II and RHIC, respectively.

In Figures 1, 2, 3, 4 and 5 we present the single differential cross-section as a function of the variables $\mathcal{V}_{\text{Exp}}$ for RHIC and LHC Run II. Our predictions are shown for LO QCD (dotted red), NLO QCD (solid black) and NLO QCD + LO QED (dashed blue), considering the default scale

choice defined in Eq. (15). In first place, we study the pion ($p_T^\pi$) and photon ($p_T^\gamma$) transverse-momentum spectrum in Fig. 1. The cross-section increases for higher c.m. energies and the impact of the QED corrections also becomes more sizable. In the case of RHIC, the distribution in $p_T^\pi$ decreases faster than the $p_T^\gamma$-spectrum, mainly due to the convolution with the FFs and the kinematical cuts. In fact, the experimental cuts imposed ensure an important contribution of events with close-to-Born kinematics. When considering LHC Run II, we appreciate that the $p_T^\pi$ spectrum falls slower than for RHIC because pions can reach higher transverse momentum. In this case, $p_T^\gamma$ is associated to the transverse momentum of the parton $c$ which fragments into a pion with momentum fraction $z$. Since the FFs tend to favour the region with $z \le 0.2$ [66], the suppression observed in Fig. 1 can be understood. Regarding the $p_T^\gamma$-spectrum, it is important to notice that different ranges for LHC Run II (left) and RHIC (right) were used, because of the very different c.m. energies and the improved capabilities of ATLAS/CMS for an efficient detection of high-energy photons.

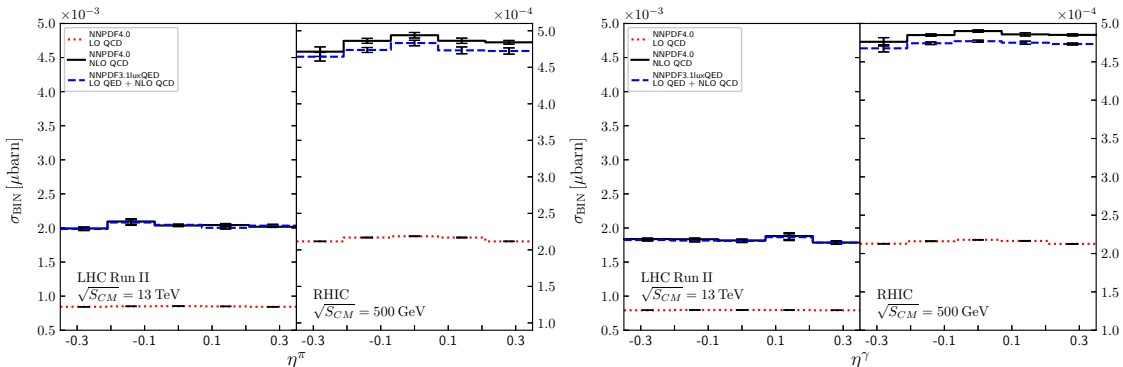

Figure 2: Same as Fig. 1, but now as a function of the rapidity of the pion (left) and the photon (right), respectively.

Next, we present the distributions in the rapidities (Fig. 2) and the azimuthal variable $\cos(\phi^\pi - \phi^\gamma)$ (Fig. 3). In both cases, we show a comparison between RHIC and LHC Run II. Even if the azimuthal range of ATLAS/CMS is wider, we restrict in these plots to the same scale, in order to provide a more fair comparison. For the rapidity distribution, we observe a significant NLO QCD correction, although the added LO QED effects are very small. Regarding the azimuthal spectrum, we can observe in Fig. 3 a peak in the back-to-back region (i.e. $\cos(\phi^\pi - \phi^\gamma) = -1$), with a fast suppression for configurations beyond Born-level kinematics.

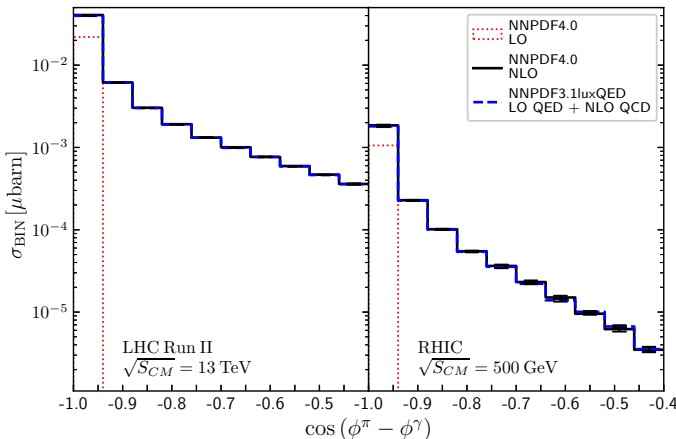

Figure 3: Dependence on $\cos(\phi^\pi - \phi^\gamma)$ for LHC Run II (left) and RHIC (right).

Besides the distributions w.r.t. the experimentally-accessible quantities, we can compute the differential cross-section as a function of the partonic momentum fractions, $x_1$, $x_2$ and $z$. For $p + p$ collisions we consider only the distributions w.r.t. $x_1$ due to the symmetry of the system. In what follows, $x$ and $x_1$ will be used interchangeably. The corresponding plots are shown in Fig. 4, for $x = x_1$ (left) and $z$ (right). In the case of RHIC, we notice that the experimental cut $p_T^\gamma \leq 15$ GeV induces a restriction on the maximum value of $x$ involved in the collision. In fact, using a LO approximation, we get

$$x_{\text{Max}} \approx \frac{p_T^\gamma}{\sqrt{S_{CM}}}, \tag{18}$$

beyond this value, the cross-section is drastically suppressed. For RHIC, we can estimate $x_{\text{Max}} \approx 0.03$. Thus, we will use this information to restrict the $x$-range in the correlation analysis presented in the next section. In this way, we will avoid dealing with regions with a negligible amount of events. In the case of LHC Run II, both for ATLAS and CMS, there is no explicit upper cut in the transverse momentum of the detected photon. However, from Fig. 1, we appreciate that the cross-section for $p_T^\gamma \geq 150$ GeV is very suppressed w.r.t. the region with $p_T^\gamma \approx \mathcal{O}(10\,\text{GeV})$. This imposes an indirect upper limit in the reconstruction of $x$, with $x_{\text{Max}} \approx 0.01$ for LHC Run II data. Since this range is smaller than in the case of RHIC, we will study the reconstruction of the partonic momentum fractions relying on simulations for RHIC experiment. Furthermore, we decided to reconstruct the partonic momentum fractions for RHIC kinematics since we can compare with previous results available in the literature [25].

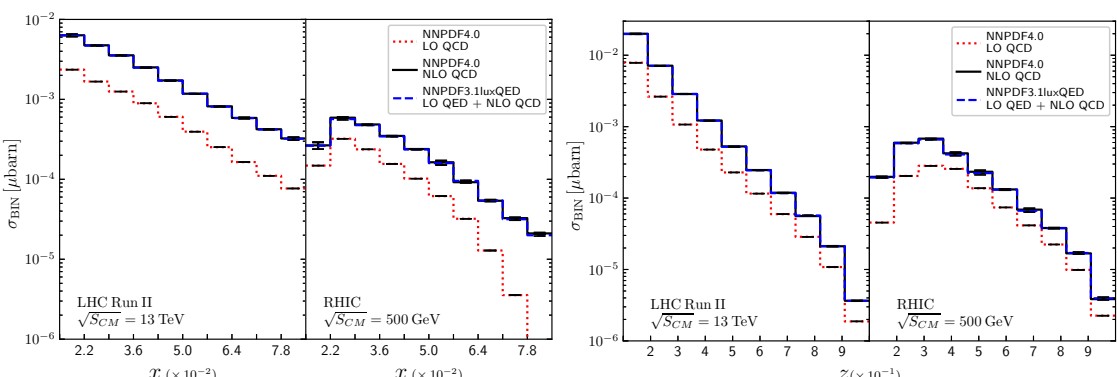

Figure 4: Cross-section as a function of the partonic momentum fractions $x$ (left) and $z$ (right), for RHIC and LHC Run II . Since these experiments involve $p + p$ collisions, we consider $x = x_1$ as given by Eq. (2).

Regarding the dependence in $z$ (right panel of Fig. 4), it reaches almost the endpoint region (i.e. $z = 1$) with a reasonable amount of events. The fact that we impose $p_T^\pi \geq 2$ GeV translates into a lower bound for $z$ given by

$$z_{\text{Min}} \approx \frac{p_T^\pi}{\sqrt{S_{CM}}}, \tag{19}$$

which corresponds to $z_{\text{Min}} \approx 0.004$ and $z_{\text{Min}} \approx 0.0001$ for RHIC and LHC Run II, respectively. Opposite to the case of the $x$-distribution, here the higher the energy of the process, the wider the accessible $z$-range. It is worthwhile noticing that the FFs used in this work do not include in the fit data with $z \leq 0.05$ and extrapolations into that region are most likely unreliable. The distribution present a peak, located at $z_{\text{Peak}} \approx 0.35$ for RHIC ($z_{\text{Peak}} \approx 0.25$ for LHC Run II). The position of the peaks depends on the explicit functional form of the PDFs and the FFs.

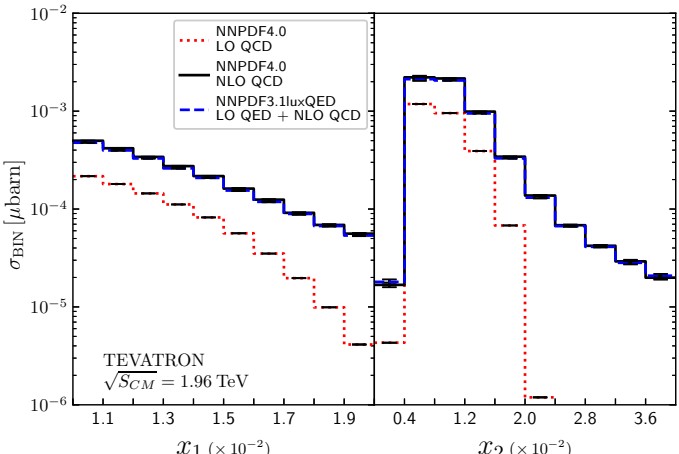

Figure 5: Cross-section as a function of the partonic momentum fraction $x_1$ (left) and $x_2$ (right), for Tevatron. $x_1$ corresponds to the momentum fraction associated to the proton, whilst $x_2$ to the antiproton.

To conclude this section, we study the case of $p + \bar{p}$ collisions at Tevatron, with $\sqrt{S_{CM}} = 1.96$ TeV. This process is of interest because it might exhibit a different dependence on PDFs and FFs, compared to $p + p$ collisions. We use the same detector cuts as in the PHENIX experiment to be able to compare with RHIC. In this case, the symmetry between $x_1$ and $x_2$ is broken, since $x_1$ ($x_2$) corresponds to the momentum fraction of a parton inside a proton (antiproton). In Fig. 5 we present the distribution for $x_1$ (left) and $x_2$ (right). We can appreciate that the distribution in $x_2$ reaches a peak around $x_2 \approx 0.01$ and then falls faster than the $x_1$-distribution. We know from previous studies that the partonic channel $gg$ is dominant [25], and thus we expect the differences to take place in the $q\bar{q}$ and $qQ$ channels. This also has an impact when studying the $x_1$ vs $x_2$ correlations, as we will show in the next subsection.

## 3.2 Correlations with the partonic momentum fractions

Since one of the main goals of this work is to reconstruct the partonic kinematics starting from experimentally accessible quantities, it is useful to first study the correlations among the different variables. This helps us to prioritize certain ansatzes depending on their functional form, in such a way that we capture the leading behaviour when exploring linear models. In the following, we restrict the discussion to RHIC kinematics (with the cuts defined in the previous section).

We start by considering the relation between $x = x_1$ and the transverse momentum of the particles in the final state. In Fig. 6, we present the correlation between $x_1$ and $p_T^\gamma$ (left column) and $p_T^\pi$ (right column). Each bin contains the corresponding integrated cross-section at LO QCD (upper row) and NLO QCD + LO QED (lower row) precision. Notice that the inclusion of higher-order corrections leads to a broadening of the patterns, originated by the presence of events in previously empty bins due to an extended phase-space. This is a general behaviour that also manifests when studying the correlations of other variables. Events with low $p_T^\gamma$ are associated with low $x_1$, and there is a somehow linear correlation between these variables. Events with low $p_T^\pi$ are mostly uniformly spread in the region of $x_1 \in [0.02, 0.6]$. This behaviour is expected from the fact that the photon originates from the partonic event (its energy is directly related to the energy of the colliding partons), whilst the pion comes from a hadronization (which implies the convolution with the FF and the consequent spreading of the distributions).

Next we move on to analyze the correlation between $x = x_1$ and the rapidities of the

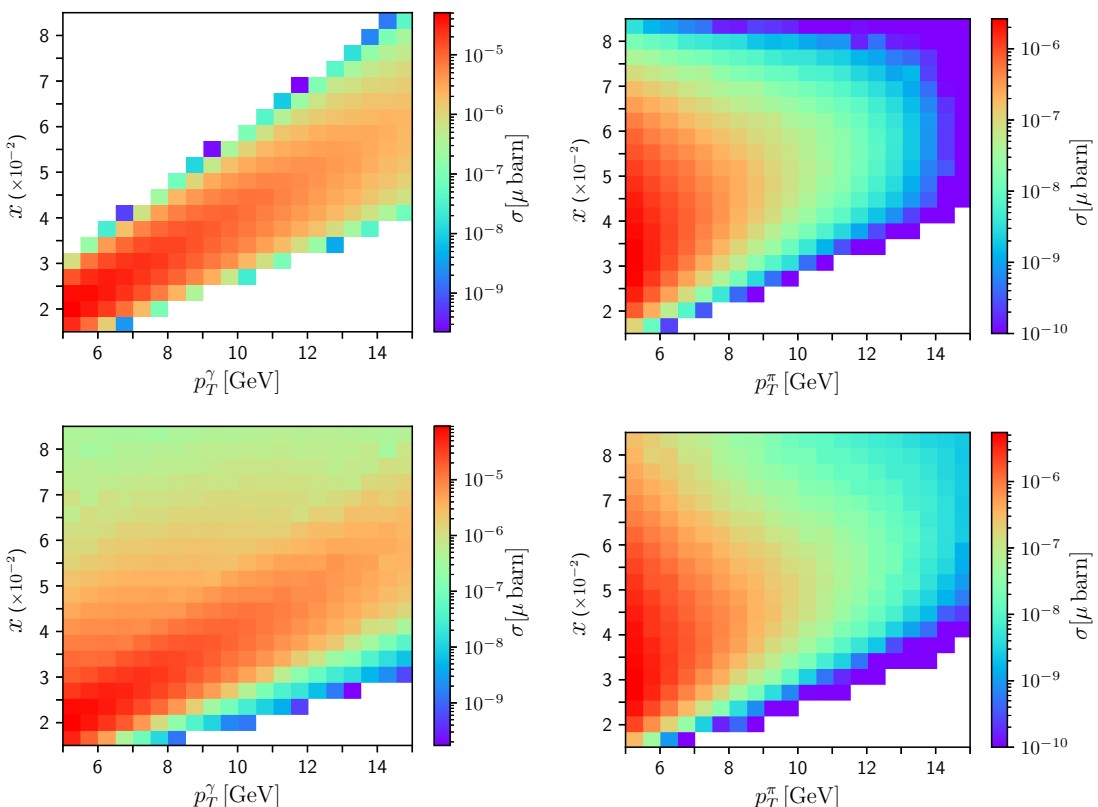

Figure 6: The partonic momentum fraction $x = x_1$ as a function of $p_T^\gamma$ (left) and $p_T^\pi$ (right). The color scale shows the cross-section at LO QCD (upper row) and NLO QCD + LO QED (lower row). We simulated the events using RHIC kinematics with $\sqrt{S_{CM}} = 500$ GeV.

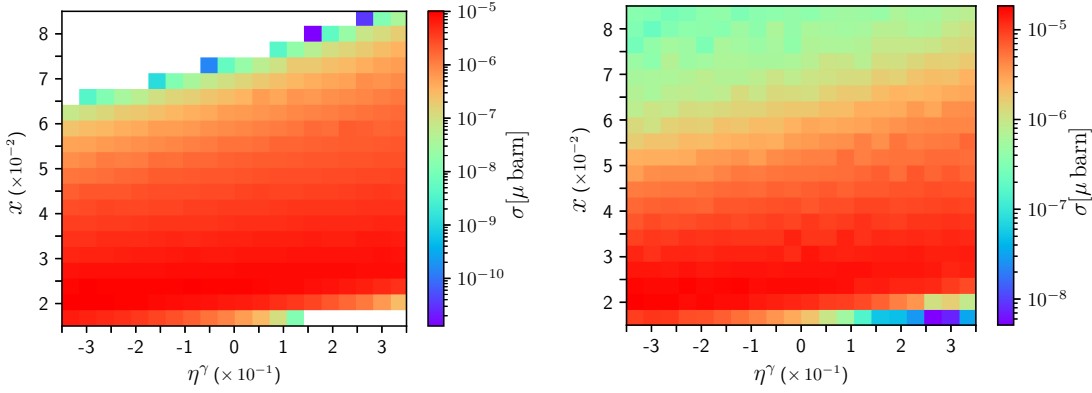

Figure 7: The partonic momentum fraction $x$ as a function of the rapidity of the photon. The color scale shows the cross-section at LO QCD (left) and NLO QCD + LO QED (right) accuracy. We simulated the events using RHIC kinematics with $\sqrt{S_{CM}} = 500$ GeV.

particles in the final state. It is important to highlight that the analysis here does depend on the momentum fraction being used, i.e. $x_1$ or $x_2$, since the rapidity introduces an asymmetry in the direction of the colliding particles. We show, in Fig. 7, the plots of $x_1$ vs. $\eta^\gamma$ at LO QCD (left) and NLO QCD + LO QED (right), respectively. Similar results were found when

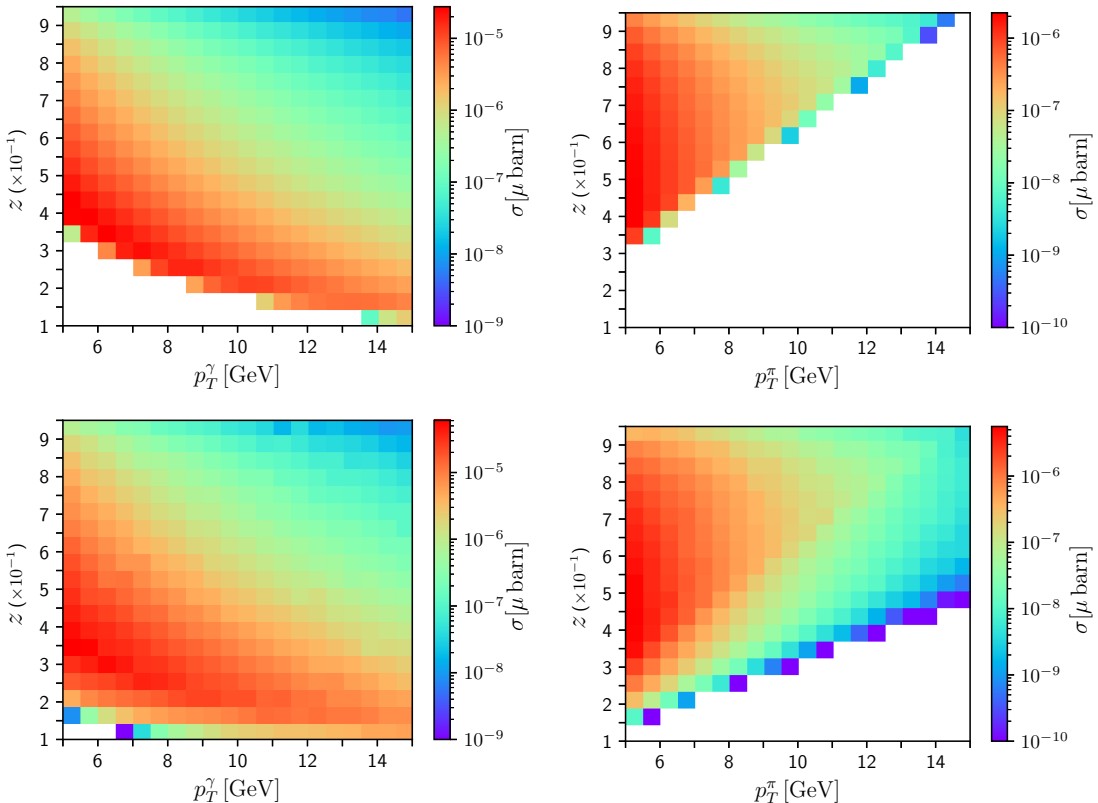

Figure 8: Partonic momentum fraction $z$ as a function of $p_T^\gamma$ (left) and $p_T^\pi$ (right). The color scale shows the cross-section at LO QCD (upper row) and NLO QCD + LO QED (lower row). We simulated the events using RHIC kinematics with $\sqrt{S_{CM}} = 500$ GeV.

considering $x_1$ vs. $\eta^\pi$ and are thus not presented here. Since the distributions are rather flat for $-0.3 \leq \eta \leq 0.3$, we find that most of the events are uniformly distributed for $x_1 \in [0.02, 0.05]$. Finally, notice that below $x_1 \approx 0.02$, the cross-section falls steeply as a consequence of the imposed kinematical cuts, and the bins are empty.

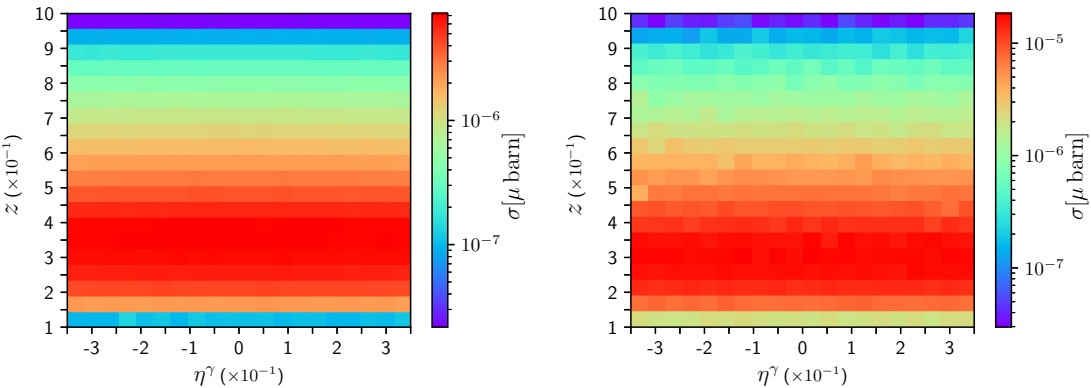

Figure 9: Partonic momentum fraction $z$ as a function of the rapidity of the photon. The color scale shows the cross-section at LO QCD (left) and NLO QCD + LO QED (right) accuracy. We simulated the events using RHIC kinematics with $\sqrt{S_{CM}} = 500$ GeV.

The analogous results on the $z$ dependence are presented in Fig. 8, the upper (lower) row corresponding to the LO QCD (NLO QCD + LO QED) contributions. On the left column we show the correlation between $z$ and $p_T^\gamma$, and between $z$ and $p_T^\pi$ on the right column. The former seems to be slightly negative, i.e. smaller values of $z$ tend to be favoured in events with higher $p_T^\gamma$, while the latter has a concentration of events in the low $p_T^\pi$ region with $z \geq 0.4$. Also, as expected, events with high $p_T^\pi$ require higher values of $z$ since the amount of partonic energy is limited by the cut $p_T^\gamma \leq 15$ GeV.

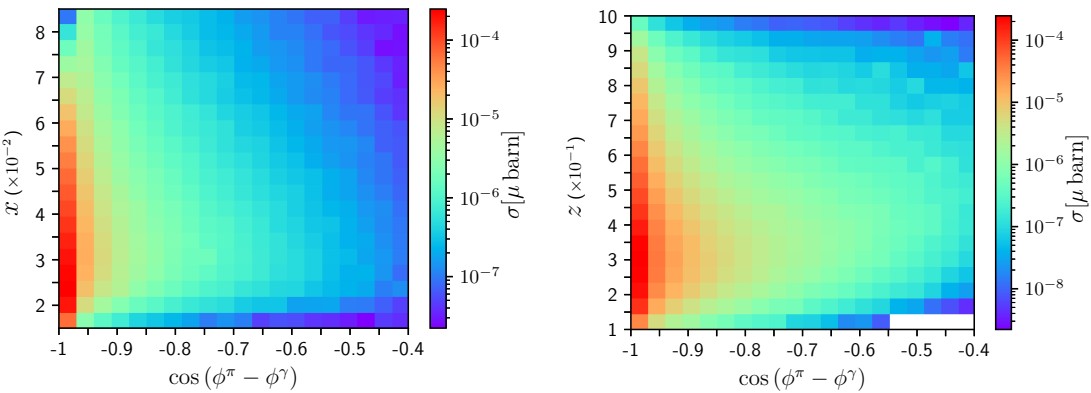

Figure 10: Partonic momentum fractions $x_1$ (left) and $z$ (right) as a function of $\cos(\phi^\pi - \phi^\gamma)$. The color scale shows the integrated cross-section value per pixel with NLO QCD + LO QED accuracy. We simulated the events using RHIC kinematics with $\sqrt{S_{CM}} = 500$ GeV.

The correlation between $z$ and the rapidities of the final state particles shows a rather flat dependence on $\eta$, as depicted in Fig. 9 for the case of $\eta^\gamma$ (similar plots were obtained when considering $\eta^\pi$).

Then, let us consider the correlations with the azimuthal variable $\cos(\phi^\pi - \phi^\gamma)$ in Fig. 10. Of course, the contributions associated to the Born kinematics are restricted to the first bin because $\cos(\phi^\pi - \phi^\gamma) = -1$ (i.e. the pion and the photon are produced back-to-back). In the remaining bins the cross-section is heavily suppressed, since it only receives contributions from real radiation (i.e. they are associated to higher-orders). We see that the events are strongly concentrated in the medium and low-$x$ region without a clear trend or dependence w.r.t. $\cos(\phi^\pi - \phi^\gamma)$. For $z$, the distribution spreads over more bins, and there is a subtle trend to favour events with a bigger azimuthal separation (smaller values of $-\cos(\phi^\pi - \phi^\gamma)$) and slightly lower values of $z$.

Finally, we analyze the correlation between $x_1$ and $x_2$ for $p + p$ collisions. In Fig. 11, we show the correlation plots at LO QCD (left) and NLO QCD + LO QED (right) accuracy, for RHIC kinematics. As expected, there is a compact region containing events at LO, reflecting the kinematical constraints of a $2 \rightarrow 2$ process. The events are concentrated in the low-$x$ region and show a strong positive linear correlation between $x_1$ and $x_2$: this reflects the fact that it is more probable to have events in the back-to-back region, in agreement with Fig. 10. When introducing higher-order corrections, the real emission phase-space gets enlarged and the distributions are spread. In any case, the positive correlation between $x_1$ and $x_2$ remains, with a strong concentration of events in the middle and low-$x$ region. Also, it is worth appreciating that the NLO real corrections are not enough to enhance the number of events with rather different values of $x_1$ and $x_2$. This is, in part, a consequence of the kinematical cuts that favour central events rather than highly boosted ones.

To conclude this section, let us comment on the importance of the study of correlations. Since we want to reconstruct the partonic momentum fractions by using the measurable vari-

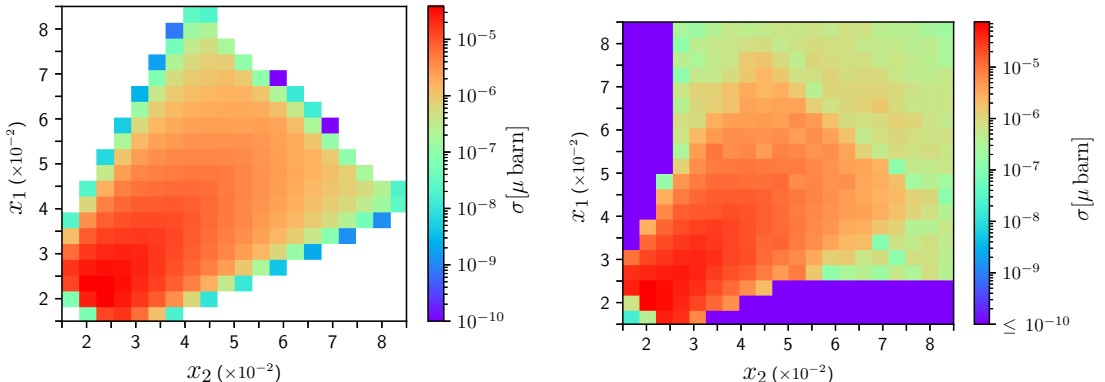

Figure 11: Correlation between $x_1$ and $x_2$ at LO QCD (left) and NLO QCD + LO QED (right) accuracy. Even when including NLO real corrections, there is a strong suppression for those events with rather different values of $x_1$ and $x_2$. We simulated the events using RHIC kinematics with $\sqrt{S_{CM}} = 500$ GeV.

ables, it is important to know which ones are the most relevant. From the previous discussion, we expect that $x$ strongly depends on $p_T^\gamma$ (positive correlation) but not on the other variables. Analogously, $z$ exhibits a negative correlation with $p_T^\gamma$, a positive one with $p_T^\pi$ and a slight dependence on $-\cos(\phi^\pi - \phi^\gamma)$. This knowledge will be applied to the construction of a basis of functions for determining $x$ and $z$ in the next section.

## 4 Reconstruction of parton kinematics

We now focus on our main goal, which is to determine the partonic variables $x_1$, $x_2$ and $z$ in terms of the measured momenta of the final state particles. At LO this is fully determined by energy-momentum conservation, and thus the LO case will serve as control. The real challenge appears at NLO, where real emissions prevent a straightforward determination of closed analytic formulae: this is what we will attempt to approximate using ML.[7]

Before entering into the details of the methodology, let us briefly mention the importance of such reconstruction in $p + p$ collisions. In electron-electron ($e + e$) collisions, the initial state is composed by fundamental particles and therefore the kinematics of the particles entering the partonic process is well-known. In the DIS processes, where a lepton collides with a nucleon, measuring the momenta of the scattered lepton provides access to the kinematics of the exchanged particles. The complexity of this scenario is higher than the $e + e$ case, but it is possible to achieve an efficient reconstruction as discussed in Ref. [4]. However, as we already explained in the Introduction, the presence of two composite colliding particles makes it non-trivial to unveil the kinematics of the fundamental objects entering in the collision. Constraining $x_1$ and $x_2$ from measurements of objects in the final state allows to understand the partonic dynamics and perform a more accurate comparison with theoretical models. We can see a clear example of this in the case of proton-nucleus collisions, when one of the PDFs has to be replaced by a nuclear PDF (nPDF). As one moves from low to high $x$, the nPDFs exhibit a pattern of suppression-enhancement w.r.t. the proton PDF (for recent nPDF studies including LHC and RHIC data, see e.g. [68]). Thus, the incorrect identification of the underlying kinematics can lead to assume that one observable is sensitive to e.g. a suppression region, when

---

[7]Doing a formal description of the ML methods that we used is beyond the scope of this work, and would take much more than a simple article. Moreover much literature is available on the topic (see e.g. [67]), so we will leave out such a discussion and mention just a few basic concepts needed in the rest of the section.

it is actually the opposite case. In turn, this could lead to an inadequate interpretation of the data or theoretical modelling of the initial state nuclear effects. Furthermore, as motivated in Refs. [25,38], it could be used to impose more tight constraints on PDF fits. For all of these, our goal is particularly relevant in the context of high-precision particle physics phenomenology.

In supervised ML, we have an initial set of data (the *training set*) and we want to map it into another known set (the *target*). Each entry in the training set is a vector of dimension $d$, with $d$ the number of variables (*features*) that the target depends upon. We also assume that there is some underlying function, the so-called *target* function, that connects the two; the task of a ML algorithm is to find a good estimation of this function. This estimator, in turn, depends on a set of parameters that is determined by minimising a function (the *cost* function) that measures some distance between the prediction of the estimator and the actual targets. As a last step, one takes another set of data with corresponding labels (*test* set) and compares how well the estimator performs for it. To prevent the estimator from performing well for the training data set but poorly for the test set (*overfitting*), the cost function includes also some parameters to control the trade off between a low training cost and a low test cost. The total number of *regularization* parameters depend on the specific method used, and the optimal value/s have to be found by picking the one/s that minimize the test cost function.

Armed with these basic concepts, we first discuss the generation of our input and target sets using the outputs of our MC code. After that, we present results obtained through the application of supervised ML for estimating $x \equiv x_1$ and $z$ at LO QCD and NLO QCD + LO QED accuracy. For the purpose of the present analysis, we explore three models: a Linear Model (LM), a Gaussian Regression (GR) and the Multi-Layer Perceptron (MLP) algorithm based on neural networks. These models have been implemented in Python using the `scikit-learn` library [69].

Before discussing the work carried out, we would like to draw attention to an important point. In the context of this analysis, *data* does not refer to *experimental data*, as this observable is yet to be measured. What the reader should interpret as *data* are the outcome of numerical simulations. Given that we are dealing with purely theoretical/phenomenological calculations, we can work without further processing the results as one would need when comparing with real data.[8]

## 4.1   Construction of the training data sets

The training and test sets were generated with the MC code used and described in the previous sections. The training set was taken to be an 80% of the full set, with randomly selected points, while the remaining data composed the test set. As was mentioned already, it deals independently with each term of the computation (LO, NLO real radiation, NLO virtual terms, NLO counter-terms). This poses two difficulties when generating the training set for feeding the ML functions. On the one hand, only the LO calculations are finite on their own; for the NLO cross-section, we have to combine all terms (real, virtual and counter-terms) to have a meaningful finite quantity. On the other hand, each term is computed through an independent MC integration. Since no two identical points are generated in the different samplings, the fully local cancellation of the divergences is spoiled. Instead, one has to split the different variables into bins and sum over all contributions entering each of them. If a sufficient number of points is sampled, the divergences cancel and we obtain the finite cross-section per bin. This is a common feature of MC integration, and many codes provide routines that take care of this for one-dimensional binning. In our case we are interested in a more differential observable, so that we had to generate a large number of points to meet this condition. Moreover, not all

---

[8]In that case, we would need to simulate the parton shower in the theoretical calculation or request the experimental data to be unfolded.

sampled points pass the selection cuts, e.g. from the $10^9$ points sampled we retain $\approx 30\%$ at LO.

For the LO we can directly use the generated points, but for the NLO case we need to discretize the differential cross-section w.r.t. the external kinematical variables defined in Eq. (17). For this purpose, we create a five-dimensional grid by binning the variables in $\mathcal{V}_{\text{Exp}}$. Explicitly, we define 10 bins for $p_T^\gamma$ and $p_T^\pi$, 5 bins for $\eta^\gamma$ and $\eta^\pi$, and 6 bins for $\cos(\phi^\pi - \phi^\gamma)$. The set of discretized experimentally-measurable variables is denoted as

$$\bar{\mathcal{V}}_{\text{Exp}} = \{\bar{p}_T^\gamma, \bar{p}_T^\pi, \bar{\eta}^\gamma, \bar{\eta}^\pi, \overline{\cos}(\phi^\pi - \phi^\gamma)\}, \tag{20}$$

where $\bar{a}$ denotes the mean value of the variable $a$ in a given bin. In total $\bar{\mathcal{V}}_{\text{Exp}}$ contains 15000 bins. Then, we define the cross-section per bin according to

$$\sigma_j(\bar{p}_T^\gamma, \bar{p}_T^\pi, \bar{\eta}^\gamma, \bar{\eta}^\pi, \overline{\cos}(\phi^\pi - \phi^\gamma)) = \int_{(p_T^\gamma)_{j,\text{MIN}}}^{(p_T^\gamma)_{j,\text{MAX}}} dp_T^\gamma \int_{(p_T^\pi)_{j,\text{MIN}}}^{(p_T^\pi)_{j,\text{MAX}}} dp_T^\pi \ldots$$
$$\times \int dx_1 dx_2 dz \, d\bar{\sigma}, \tag{21}$$

with $x_{j,\text{MIN}}$ ($x_{j,\text{MAX}}$) the minimum (maximum) value of the variable $x$ in the $j$-th bin, $\bar{x}$ the corresponding average of $x$ over the $j$-th bin and

$$d\bar{\sigma} = \frac{d\sigma}{d\mathcal{V}_{\text{Exp}} \, dx_1 dx_2 dz}, \tag{22}$$

is the fully-differential *hadronic* cross-section as a function of the partonic momentum fractions and the experimentally-measurable variables $\mathcal{V}_{\text{Exp}}$. At LO, $\sigma_j$ can be straightforwardly calculated since we only need to integrate the tree-level scattering amplitude in a $2 \to 2$ phase-space. However, as we explained in Sec. 2, the NLO corrections include several contributions calculated with different kinematics (virtual, real, counter-terms): all of these are taken into account in $d\bar{\sigma}$ and integrated over their corresponding phase-space to obtain $\sigma_j$.[9]

Once the grid and the discretized cross-section are defined, we use the MC code to generate three histograms per bin in the grid. These histograms corresponds to the distributions $d\sigma_j/dx_1$, $d\sigma_j/dx_2$ and $d\sigma_j/dz$, respectively. So, given a point in the grid

$$p_j = \{\bar{p}_T^\gamma, \bar{p}_T^\pi, \bar{\eta}^\gamma, \bar{\eta}^\pi, \overline{\cos}(\phi^\pi - \phi^\gamma)\} \in \bar{\mathcal{V}}_{\text{Exp}}, \tag{23}$$

we can define

$$(x_1)_j = \sum_i (x_1)_i \frac{d\sigma_j}{dx_1}(p_j; (x_1)_i), \tag{24}$$

$$(x_2)_j = \sum_i (x_2)_i \frac{d\sigma_j}{dx_2}(p_j; (x_2)_i), \tag{25}$$

$$(z)_j = \sum_i z_i \frac{d\sigma_j}{dz}(p_j; z_i), \tag{26}$$

which correspond to the weighted average of the partonic momentum fractions extracted from the histograms generated with the MC code. This is done because, given a certain final state, the partonic momentum fractions are not unambiguously defined. Beyond LO, the

---

[9]Binning could be avoided using a fully-local framework for computing higher-order corrections [70, 71]. One of these methods is the Four-Dimensional Unsubtraction (FDU) [72–75] based on the Loop-Tree Duality [76–78]. Since FDU leads to a fully-differential and finite representation of the cross-section, it constitutes a perfectly suited candidate to improve the efficiency of the analysis presented in this article.

real-radiation contributions include processes that contain extra-particles in the final state. As a consequence, the momentum conservation equations for each event lead to a non-unique value for the momentum fractions. Thus, in Eqs. (24)-(26), we define an *equivalent* event using the fully-differential cross-section to weight the contributions of the different partonic configurations.

At this stage, we can identify $\bar{\mathcal{V}}_{\text{Exp}}$ as the training set and $\{(x_1)_j, (x_2)_j, (z)_j\}$ as the target one. Then, we can train the ML algorithms to find the target functions

$$X_{1,\text{REC}} := \bar{\mathcal{V}}_{\text{Exp}} \longrightarrow \bar{X}_{1,REAL} = \{(x_1)_j\}, \tag{27}$$

$$X_{2,\text{REC}} := \bar{\mathcal{V}}_{\text{Exp}} \longrightarrow \bar{X}_{2,REAL} = \{(x_2)_j\}, \tag{28}$$

$$Z_{\text{REC}} := \bar{\mathcal{V}}_{\text{Exp}} \longrightarrow \bar{Z}_{REAL} = \{(z)_j\}, \tag{29}$$

that will allow us to reconstruct the MC partonic momentum fractions $\bar{X}_{1,REAL}$, $\bar{X}_{2,REAL}$ and $\bar{Z}_{REAL}$.

To conclude this discussion, notice that the definitions given in Eqs. (24)-(26) are crucial beyond LO. In fact, for a $2 \to 2$ process, fixing the bin $p_j \in \bar{\mathcal{V}}_{\text{Exp}}$ leads to a unique value of the partonic-momentum fractions. Explicitly, we have

$$X_{1,\text{REC}} = \frac{p_T^\gamma \exp(\eta^\pi) + p_T^\gamma \exp(\eta^\gamma)}{\sqrt{S_{CM}}}, \tag{30}$$

$$X_{2,\text{REC}} = \frac{p_T^\gamma \exp(-\eta^\pi) + p_T^\gamma \exp(-\eta^\gamma)}{\sqrt{S_{CM}}}, \tag{31}$$

$$Z_{\text{REC}} = \frac{p_T^\pi}{p_T^\gamma}, \tag{32}$$

as explained in Ref. [25]. Due to the presence of $2 \to 3$ sub-processes contributing to the real radiation, the value of $\{x_1, x_2, z\}$ for a given $p_j$ is not unambiguously defined at NLO (and beyond). If we pick an event with a fixed $p_j$ from our NLO MC generator, the real partonic momentum fractions might take all the kinematically-allowed values. However, the probability of the different outcomes is given by the differential-cross section of the event, which motivates the definitions introduced in Eqs. (24)-(26). In the following, we explain how these data sets are used with the different ML frameworks.

## 4.2 Linear regression

Linear methods, as the name indicates, provide the estimation of the target function as a linear combination of the input set. However, the linearity occurs at the level of the parameters and one can apply prior knowledge to construct new features upon which the target dependence is simpler. Choosing a *good* set of features (basis) plays an important role to achieve an accurate reconstruction.

For example, at LO we take inspiration from the exact analytical expressions given by Eqs. (30)-(32) and propose the basis

$$\mathcal{B}_{\text{LO}} = \{\frac{p_T^\gamma}{\sqrt{S_{CM}}} \exp(\eta^\pi), \quad \frac{p_T^\gamma}{\sqrt{S_{CM}}} \exp(\eta^\gamma), \quad \frac{p_T^\gamma}{\sqrt{S_{CM}}} \exp(-\eta^\pi), \quad \frac{p_T^\gamma}{\sqrt{S_{CM}}} \exp(-\eta^\gamma), \quad p_T^\pi/p_T^\gamma\}. \tag{33}$$

We then expect $x_1$ to be well reconstructed by a linear combination of the first two elements of the basis (with coefficient 1), whilst $z$ should be mainly proportional to the last element. In Fig. 12, we show the correlation between the MC partonic momentum fractions (vertical axis) and the output of the linear regression (horizontal axis). Each bin contains the integrated

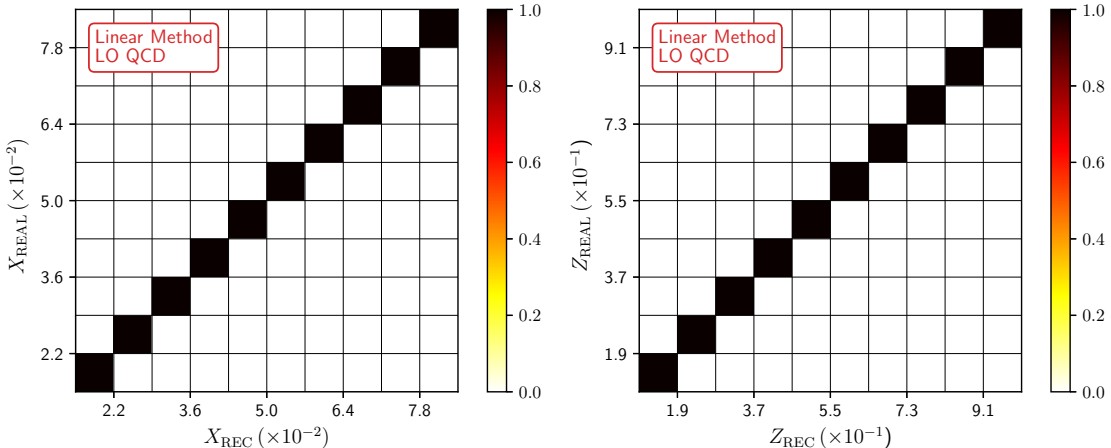

Figure 12: Correlation between the MC momentum fractions (i.e. $X_{\text{REAL}}$ and $Z_{\text{REAL}}$) and the ones obtained at LO QCD accuracy using the LM approach. Each bin of the correlation plot is filled with the integrated cross-section.

cross-section at LO QCD accuracy. The reconstruction is perfect, and the LM approach leads exactly to the Eqs. (30)-(32).

When dealing with the NLO scenario, in principle, we should expect an enlargement of the basis. The elements of $\mathcal{B}_{\text{LO}}$ are not enough to fully capture the additional dependencies introduced by the NLO real kinematics. In fact, in Ref. [25] the authors proposed

$$X_{1,\text{REC}} = \frac{p_T^\gamma \exp(\eta^\pi) - \cos(\phi^\pi - \phi^\gamma) p_T^\gamma \exp(\eta^\gamma)}{\sqrt{S_{CM}}}, \tag{34}$$

$$X_{2,\text{REC}} = \frac{p_T^\gamma \exp(-\eta^\pi) - \cos(\phi^\pi - \phi^\gamma) p_T^\gamma \exp(-\eta^\gamma)}{\sqrt{S_{CM}}}, \tag{35}$$

$$Z_{\text{REC}} = -\cos(\phi^\pi - \phi^\gamma) \frac{p_T^\pi}{p_T^\gamma}, \tag{36}$$

that agree with Eqs. (30)-(32) at LO, but introduce an additional dependence on the azimuthal variables at higher-orders. The study of correlations performed at NLO QCD accuracy using these expressions showed a good reconstruction of the MC partonic momentum fractions.

With this precedent in mind, we propose here to include additional functional dependencies to have a more flexible reconstruction. We start by defining a primitive set of functions

$$\mathcal{K} = \{\frac{p_T^\gamma}{\sqrt{S_{CM}}}, \frac{p_T^\pi}{\sqrt{S_{CM}}}, \exp(\eta^\gamma), \exp(\eta^\pi), \cos(\phi^\pi - \phi^\gamma)\}, \tag{37}$$

in such a way that the reconstructed variables take the form

$$Y_{\text{REC}} = \sum_{i=1, i\neq 5}^{9} (a_i^Y + b_i^Y \mathcal{K}_5)\mathcal{K}_i + \sum_{i\leq j, \{i,j\}\neq 5, j-i\neq 5} (c_{ij}^Y + d_{ij}^Y \mathcal{K}_5)\mathcal{K}_i \mathcal{K}_j, \tag{38}$$

with $Y_{\text{REC}} = \{X_{1,\text{REC}}, X_{2,\text{REC}}, Z_{\text{REC}}\}$ and $\mathcal{K}_i = \mathcal{K}_{i-5}^{-1}$ for $i = \{6, 7, 8, 9\}$. The ansatz proposed in Eq. (38) generalizes the basis $\mathcal{B}_{\text{LO}}$ and includes products of up to three kinematical variables, which gives more flexibility to fit the data. In total, there are eighty-one functions in the basis, that we denominate *general basis*. However, as we will now explicitly see, a larger basis does not imply a better reconstruction.

Figure 13: Correlation between the MC momentum fractions (i.e. $X_{\mathrm{REAL}}$ and $Z_{\mathrm{REAL}}$) and the ones obtained at NLO QCD + LO QED accuracy using the LM approach ($X_{\mathrm{REC}}$ and $Z_{\mathrm{REC}}$). Upper row: using the general basis given in Eq. (38). Middle row: *physically motivated* basis. Lower row: *LO-inspired* basis.

If we take Eq. (38), with $Y = \{x_1, z\}$ we obtain the results shown in the upper row of Fig. 13. In this figure, we indicate the strength of the correlation with the integrated cross-section per bin at NLO QCD + LO QED accuracy. The coefficients $a_i^Y$, $b_{ij}^Y$, $c_{ij}^Y$ and $d_{ij}^Y$ are given in App. B. The reconstruction is good in the low-$x$ and low-$z$ region. This is expected because

the cross-section is larger in that region, so there are more data-points to perform the fit. However, the reconstruction becomes noisy and imprecise for higher values of the momentum fractions. The LM is unable to keep the functional dependencies that better approximate the real momentum fractions in regions with low number of events.

For this reason, we explore a second approach. We profit from the findings in Sec. 3.2, and distinguish different basis for $Y = x_1$ and $Y = z$. It was shown that $x_1$ exhibits a positive correlation with $p_T^\gamma$, so we remove the contributions involving $\mathcal{K}_6 = (p_T^\gamma)^{-1}$ from Eq. (38). Regarding $z$, the conclusion of Sec. 3.2 was that it is correlated with $\mathcal{K}_6 = (p_T^\gamma)^{-1}$, $\mathcal{K}_2 = p_T^\pi$ and that also presents a mild correlation with $\mathcal{K}_5$. So, we remove the contributions that involve the primitive functions $\mathcal{K}_1$ and $\mathcal{K}_7$. As a result, we propose a *physically-motivated* reconstruction by taking Eq. (38) and setting

$$
\begin{aligned}
b_6^{X_1} &= 0\,, \\
c_{6,j}^{X_1} &= d_{6,j}^{X_1} = c_{i,6}^{X_1} = d_{i,6}^{X_1} = 0 \quad \{i,j\} \in \{1,\dots,9\}\,,
\end{aligned}
\tag{39}
$$

for $x_1$ and

$$
\begin{aligned}
b_1^Z &= b_7^Z = 0\,, \\
c_{1,j}^Z &= d_{1,j}^Z = 0 \quad j \in \{1,\dots,9\}, j \neq \{5,7\}\,, \\
c_{i,7}^Z &= d_{i,7}^Z = 0 \quad i \in \{1,\dots,9\}, i \neq \{1,5\}\,,
\end{aligned}
\tag{40}
$$

for $z$. The coefficients obtained with these assumptions are presented in App. B, whilst the corresponding correlations with the real MC momentum fractions are shown in the middle row of Fig. 13. The correlation is slightly better for $z$, but it is worse for $x$. Even if the *physically-motivated* basis includes elements that are selected according to the correlations with physical variables, it turns out that the abundance of points in a particular region of the parameter space imposes a very tight constraint on the whole fit. For $z$, this is not a big problem since it seems to be dominated by the ratio $p_T^\pi/p_T^\gamma$. However, the dependence of $x$ w.r.t. the kinematical variables is more complicated, and a linear fit is not enough to capture it. Thus, reducing the basis does not lead to an improved reconstruction of the momentum fractions.

To conclude this discussion, let us mention that we tested the LM with another basis inspired by the LO formulae. Namely, this *LO-inspired* basis is given by

$$
\begin{aligned}
\mathcal{B}_{\text{NLO}}^{X_1} = \{ &\frac{p_T^\gamma}{\sqrt{S_{CM}}}\exp(\eta^\gamma), \frac{p_T^\gamma}{\sqrt{S_{CM}}}\exp(\eta^\pi), \frac{p_T^\pi}{\sqrt{S_{CM}}}\exp(\eta^\gamma), \frac{p_T^\pi}{\sqrt{S_{CM}}}\exp(\eta^\pi), \\
&\frac{p_T^\gamma \mathcal{K}_5}{\sqrt{S_{CM}}}\exp(\eta^\gamma), \frac{p_T^\gamma \mathcal{K}_5}{\sqrt{S_{CM}}}\exp(\eta^\pi), \frac{p_T^\pi \mathcal{K}_5}{\sqrt{S_{CM}}}\exp(\eta^\gamma), \frac{p_T^\pi \mathcal{K}_5}{\sqrt{S_{CM}}}\exp(\eta^\pi) \}\,,
\end{aligned}
\tag{41}
$$

for $x \equiv x_1$ and

$$
\mathcal{B}_{\text{NLO}}^Z = \{ p_T^\pi/p_T^\gamma, \mathcal{K}_5\, p_T^\pi/p_T^\gamma, \mathcal{K}_5\, p_T^\pi/\sqrt{S_{CM}}, \mathcal{K}_5\, \sqrt{S_{CM}}/p_T^\gamma \}\,,
\tag{42}
$$

for $z$. In this case, the reconstruction was even worse, as can be seen in the lower row of Fig. 13. In particular, $X_{1,\text{REC}}$ seems to be uncorrelated with $X_{1,\text{REAL}}$. So, we can appreciate that the approach followed in Ref. [25] was more efficient than the LM. In other words, forcing a linear combination that describes the LO kinematics and then using the same formulae for higher-orders, allows to achieve a more precise reconstruction. In the next subsections, we explore other methods that will lead to a better approximation of the MC momentum fractions in a more automatized way.

## 4.3 Gaussian regression

While the LM method provides a good description for the LO case, at NLO the result strongly depends on the variables used to feed the algorithm. As the larger basis seems to render a slightly better reconstruction, we could use this as a motivation to further expand our basis, e.g. by including higher-powers of its elements. However this relies on deciding *i)* which appropriate combinations of $\mathcal{K}_i$ are needed, and *ii)* to which power it would be convenient to go. The first point was addressed in Subsec. 4.2 by constructing several bases, with different degree of success. Regarding the second point, we could try with different powers of a given basis, but this would be a cumbersome task. A more general and computationally efficient approach can be implemented by using the kernel trick (see e.g. [79, 80]). In this method, the feature vector in the calculation is replaced by writing everything in terms of a function (kernel) of the dot product of the elements of the training set. In particular we use the radial basis function (RBF), defined as

$$k(x_i, x_j) = \exp\left(-\frac{d(x_i, x_j)}{2l^2}\right), \qquad (43)$$

where $x_i$, $x_j$ are two elements of the training set, $d(x_i, x_j)$ is the Euclidean distance between them, and $l$ is a distance parameter (not necessarily the same for all $\{i, j\}$). The RBF has the advantage of including all possible powers of the exponent, and therefore we expect a better reconstruction of the kinematic variables. Similarly to the LM, the GR requires a set of input variables. In order to properly compare the methods, we take the same bases for both. The GR also needs the user to select the *width* of each Gaussian function, $l$, which is by default $l = 1$. In principle it could be different for each feature of the input set, but for simplicity we keep it feature-independent. However we did find better reconstructions when using different $l$ for $x$ and $z$. The optimal values of $l$ for each basis can be found in Table 1.

Table 1: Values of the $l$-parameter to reconstruct the $x$ and $z$ momentum fractions in three different basis used within the GR framework.

| Reconstructed quantity | General basis | Physically-motivated basis | LO-inspired basis |
|:---:|:---:|:---:|:---:|
| $x$ | 26 | 30 | 1 |
| $z$ | 21 | 25 | 1.5 |

We find that, when using the most general basis, a better agreement between the reconstructed and the real data sets requires *wide* Gaussian functions. In addition, if we reduce the basis the GR tends to require *wider* Gaussian functions to achieve a good description of the data sets. Finally, we find that in the *physically-motivated* basis, the GR finds the best agreement by choosing $l = 30$ for the prediction of $x$ and $l = 25$ for $z$, i.e. *sharp* Gaussian functions are needed meaning that a combination of these variables is enough to reproduce the full data sets.

These facts can appreciated in Fig. 14 where we present the results obtained at NLO QCD + LO QED accuracy. As expected, the inclusion of higher-order terms (higher non-linearity) in the training set brings a significant improvement with respect to the LM, in particular for the reconstruction of $x$. In addition, we point out that among the three basis, in general, the reconstruction of $x$ is harder than the $z$ momentum fraction. The general basis can extract the information to almost determine completely a function for the prediction of the momentum fractions but with wide Gaussian functions. In contrast, the *physically-motivated* basis makes a good job in the determination of $z$ but is not that accurate on the extraction of $x$, although it requires *sharp* Gaussian functions, meaning that they are well localized and determined.

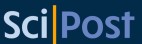

**Figure 14:** Correlation between the MC momentum fractions (i.e. $X_{\text{REAL}}$ and $Z_{\text{REAL}}$) versus the ones obtained at NLO QCD + LO QED accuracy ($X_{\text{REC}}$ and $Z_{\text{REC}}$). We show the results corresponding to the GR approach, using the general basis (upper row), the *physically-motivated* basis (middle row) and the *LO-inspired* basis (lower row).

To conclude this section, we appreciate that the GR method leads to a more reliable reconstruction of the MC momentum fractions, compared to the LM. The best results are obtained with a larger basis, in order to have more flexibility. Moreover, the non-linearity inherent to the GR allows to overcome the limitation of the overfitting in the low-*x* and low-*x* region that we observed in the LM, leading to a very accurate reconstruction in a wider range.

## 4.4 Neural Networks

Before jumping into the results of this section, let us briefly remind the reader of what is a neural network (NN). The mathematical formulation of artificial neural networks (ANN), simply known as NN nowadays, was presented more than 70 years ago [81]. Inspired in the real biological systems, ANN are a collection of connected nodes, which can transmit an activation signal from one node to the other, thus emulating the behaviour of neurons.

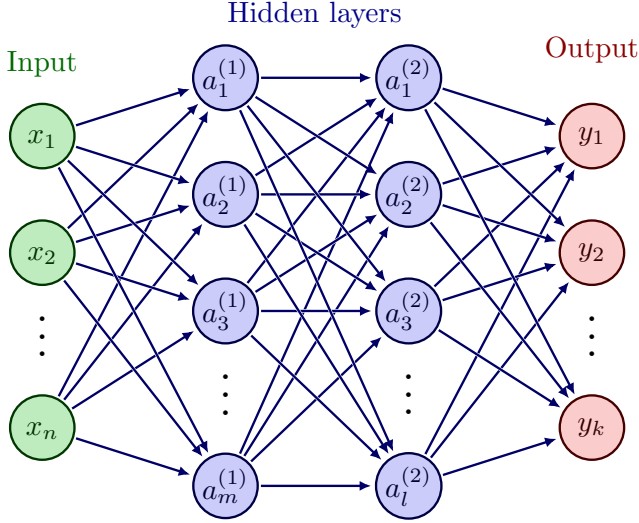

Figure 15: Graphical representation of a generic neural network (or *artificial neural network*) architecture. Between the *input* (X) and *output* (Y) layers, additional neurons can be added, organized in *hidden layers*. The arrows represent the so-called *activation functions*, that connect the different layers of the system.

From the computational point of view, the building blocks of a NN are algorithms (called *Perceptrons*) used in supervised learning to decide if an input belongs into a class or not (binary classifier). They consist of a set of input values $X$, that will be linearly combined by weights ($W$) and independent terms $B$ (*biases*), after which the sum will be transformed by the (usually non-linear) activation function $f$, giving an output $Y$: $Y = f(z)$ with $z = X * W + B$. Each Perceptron mimics a neuron, and a combination of them makes a NN. The standard nomenclature labels the inputs and outputs as input and output *layers*, respectively. To increase the capabilities of the NN (and its complexity) one can add more neurons in between, organised in *hidden* layers. The activation functions connecting one layer to the next do not need to be the same, neither the number of neurons in each hidden layer. A graphical representation of this architecture is displayed in Fig. 15. The learning proceeds in two steps. First, the NN computes the output from the inputs (feed-forward). In a second step (back-propagation), it calculates the cost and then minimizes it. This can be implemented in different ways, one of the most popular being stochastic gradient descent.[10]

The choice of the activation function/s and relevant parameters is highly non-trivial, and trial-and-error was required to find a configuration that could reproduce the momentum fractions. A non-exhaustive comparison of different combinations is presented in App. C, but here we limit ourselves to present the results corresponding to the parameters summarised in Table 2, which are used within the `scikit-learn` framework.

---

[10]This procedure depends on the size of a parameter called the *learning rate*, that also requires adjustment. For more details about the implementation of NN in `scikit-learn` and specifics of the MLP algorithm we refer the reader to Ref. [69].

Table 2: Architecture for the MLP best fit parameters for the reconstruction of the momentum fractions at LO in QCD: $X_{REC}$(LO) and $Z_{REC}$(LO) (second and third columns), and for the momentum fractions at NLO QCD + LO QED: $X_{REC}$(NLO) and $Z_{REC}$(NLO) (fourth and fifth columns).

|  | $X_{REC}$ (LO) | $Z_{REC}$ (LO) | $X_{REC}$ (NLO) | $Z_{REC}$ (NLO) |
|---|---|---|---|---|
| # of hidden layers | 2 | 1 | 5 | 5 |
| # of neurons/layer | 200 | 100 | 300 | 300 |
| activation function | ReLU | ReLU | ReLU | ReLU |
| # iterations | $1 \times 10^5$ | $1 \times 10^5$ | $1 \times 10^{12}$ | $1 \times 10^{12}$ |
| learning rate | $1 \times 10^{-3}$ | $1 \times 10^{-3}$ | $1 \times 10^{-4}$ | $1 \times 10^{-4}$ |

Regarding the cost/loss function, we rely on the default minimization strategy implemented in present in `scikit-learn`. Explicitly, it defines a *score function* which should be maximized to identify the optimal fit (thus, it is related to the *inverse* of the cost function). By default, the MLP method uses the Pearson correlation coefficient as score function. Then, the maximum is estimated by the stochastic gradient descent method [69].

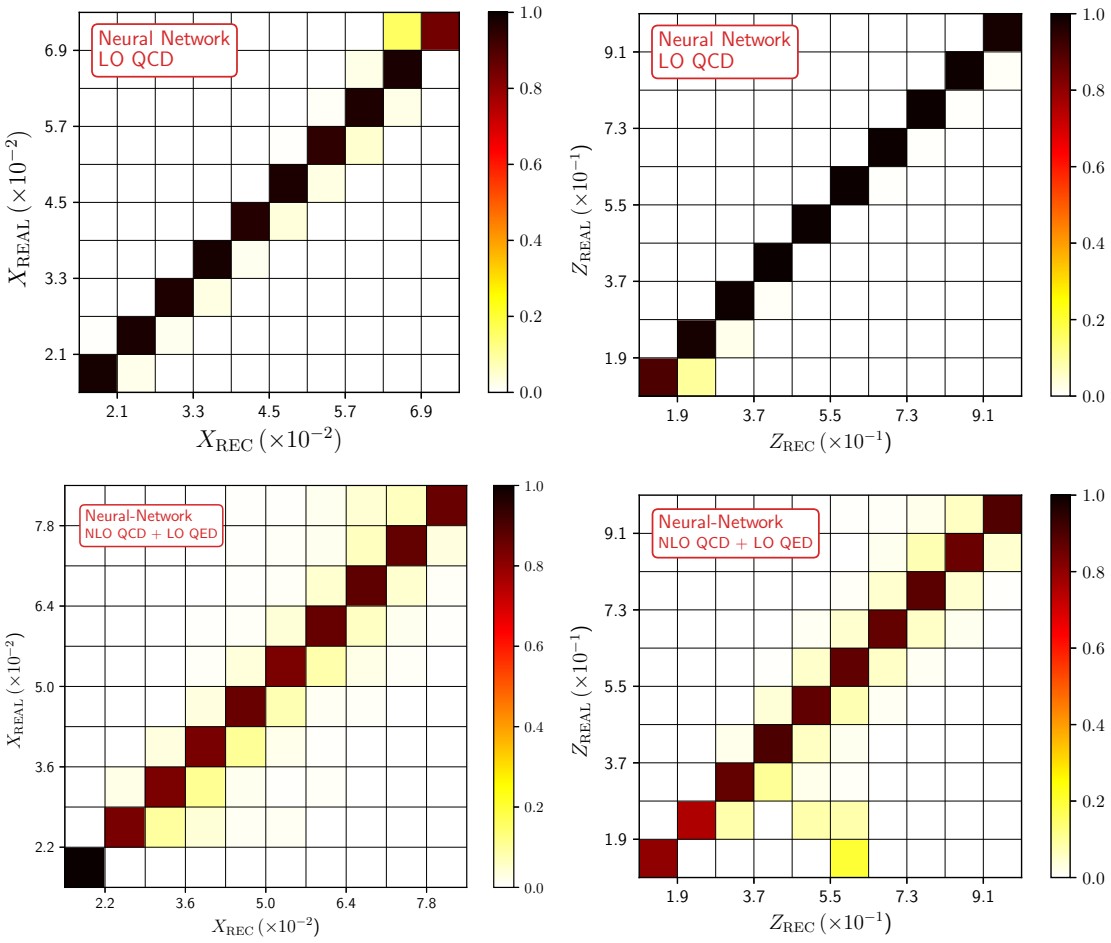

Figure 16: Left: Comparison of the momentum fractions $X_{REAL}$ and $X_{REC}$ obtained with MLP neural networks with the parameters given in Table 2. The upper (lower) row corresponds to the LO QCD (NLO QCD + LO QED) data set. Right: same as the l.h.s but for $Z_{REAL}$ and $Z_{REC}$.

The results of the MLP algorithm are presented in Fig.16 for the LO QCD contribution (upper row) and the NLO QCD + LO QED correction (lower row). In the LO case the reconstruction is quite good, without reaching the level of accuracy of the LM or GR. This is a strong evidence that the complexity of the NN machinery greatly exceeds that of the task to be solved. In the NLO case, on the contrary, the reconstruction is much better than the one obtained with the LM using any basis, and similar to the GR one with the general basis (upper row of Fig. 14). The plots show an almost perfect agreement in all bins for both $x$ and $z$. The largest discrepancy appears for $x$, which can be partially due to the higher complexity of the target function for $x$ than for $z$, already suggested by the analytic LO expressions. Indeed, almost all trials performed with different methods and configurations arrive to reasonable relations between $Z_{REAL}$ and $Z_{REC}$. However for $x$, we have to either increase the number of elements in our basis (GR) or the number of layers/nodes (NN).

In any case, we can highlight that the MLP algorithm does not require to choose any particular basis: the complexity is translated into defining the proper architecture. This task is more suitable for automation, thus more appropriate for tackling generic physical processes regardless of the number or kind of particles involved. Whereas LM or GR could take advantage from physically-motivated parameter's choice to speed-up an accurate reconstruction, the NN framework relies mainly on computational power to reduce the problem to a *black-box* function.

## 4.5 Error propagation in the reconstruction

Since all the ingredients involved in the calculation of cross-sections have associated errors, these are expected to propagate and affect the accuracy of the partonic momentum fraction reconstruction. Of course, also the regression and MLP algorithms introduce errors in the definition of $X_{REC}$. In this section we briefly describe the strategies adopted to provide a quantitative estimation of the errors, and present our results. The methodologies are explained in detail in App. A.

We started by considering the default dataset obtained by setting $\mu_R = \mu_F = \mu_I = \xi\mu$ with $\xi = 1$ and $\mu$ given by Eq. (15). From this set, we obtained the reconstructed partonic momenta $X_{REC}^{\xi=1}$, for $X = \{x_1, z\}$ and using the different reconstruction techniques (LM, GR and MLP) described along this Section. Then, we took the different datasets $\mathcal{D}^{(\xi=1/2,1,2)}$, computed the reconstructed momentum fractions and created histograms weighting each event (i.e. *datapoint*) with the integrated cross-section per bin. The results are shown in Fig. 17, where we plot the distribution in $X_{REC}$ and $Z_{REC}$ in the left and right columns, respectively. The first row corresponds to the reconstruction using the linear method (LM), the middle one uses Gaussian regression (GR) and the last one relies on MLP. We notice that, on average, the width of the band around the central value of the distributions is $\mathcal{O}(50\%)$. This is completely expected since the NLO QCD $K$-factor for this process is also $\mathcal{O}(50\%)$, as reported in Ref. [25].

Finally, we applied the second strategy explained in App. A. In this case, we trained different reconstruction functions using the different datasets $\mathcal{D}^{(\xi=1/2,1,2)}$. Then, we selected the default case (i.e. $\xi = 1$) and evaluated the reconstructed functions: we define the expected value and the error according to Eq. (51). The last step consisted in calculating the average relative error, i.e. $\Delta X(p_j)/X(p_j)$, for $X = \{x_1, z\}$. For case of the MLP method, we found an average error of 7.5% and 5.4% for the reconstruction of $x_1$ and $z$, respectively. Of course, the error depends on the kinematic region that we are studying, which implies a dependence on the value of $X_{REAL}$ that we are reconstructing. Thus, we report in Fig. 18 the correlation error bands for the reconstruction of $x_1$ (left) and $z$ (right). In the $y$-axis, we show the reconstructed momentum fraction, whilst in the $x$-axis we plot the real momentum fraction. We considered the three machine-learning methodologies: we found that the error band is

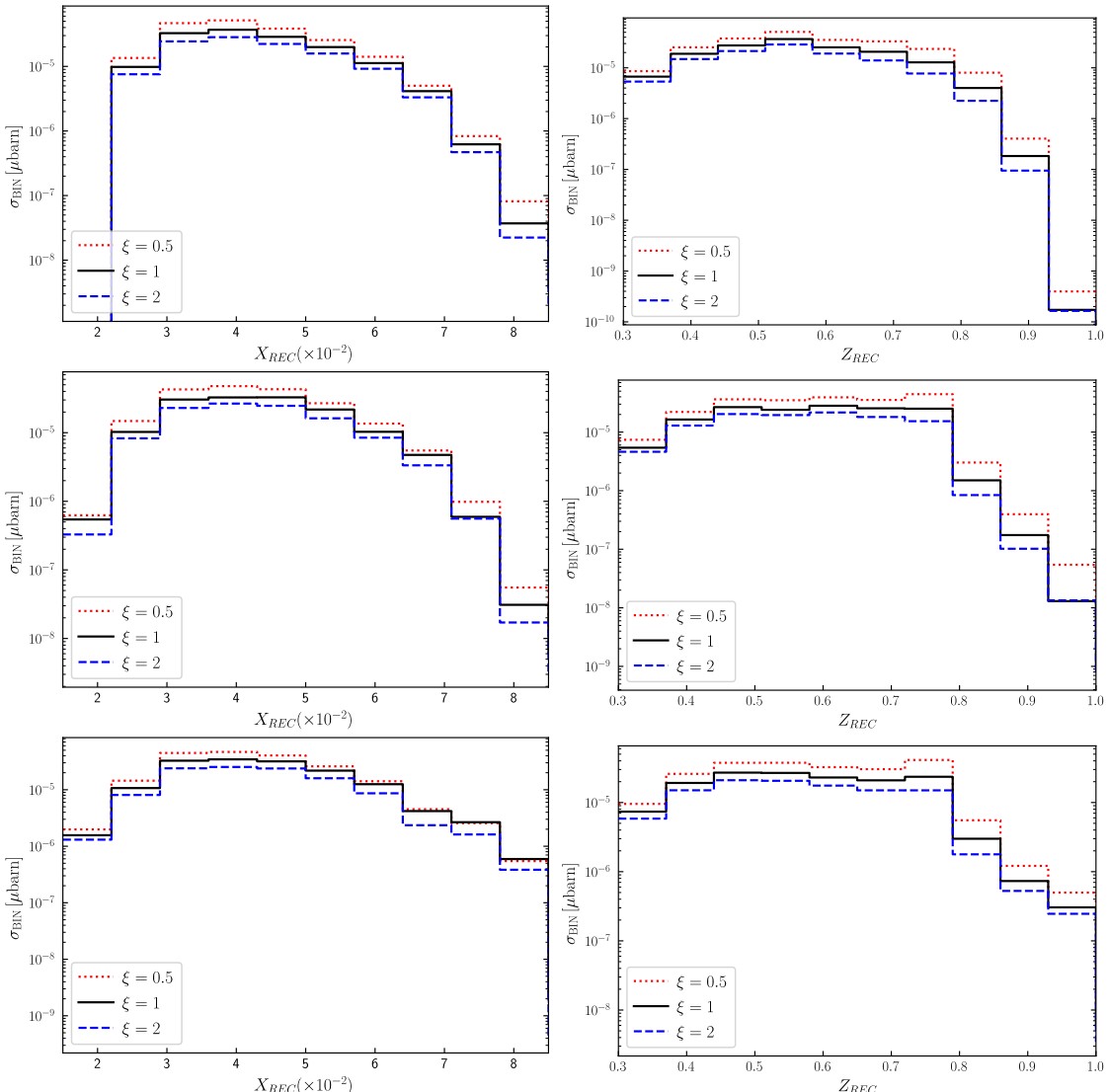

Figure 17: Histograms based on the datasets $\mathcal{D}^{(\xi=1/2,1,2)}$, showing the distributions in $X_{\text{REC}}$ (left column) and $Z_{\text{REC}}$ (right column). Each datapoint is weighted using the integrated cross-section per bin. We explored the three reconstruction methods mentioned in Sec. 4: linear method (first row), Gaussian regression (second row) and neural networks (third row).

narrower for the NN estimation in the case of $x_1$ reconstruction, but the linear method gives a more stable result for reconstructing $z$. As explained previously, thus is due to a simpler functional dependence for $z$, which is mainly dominated by the ratio $p_T^\pi/p_T^\gamma$. This effect is also present in the additional correlation plots discussed in App. A (see Fig. 19).

To conclude this discussion, the second strategy provides a global error estimation with more realistic predictions, since the first strategy drastically overestimates the reconstruction errors by fully propagating the scale uncertainties of the original observable.

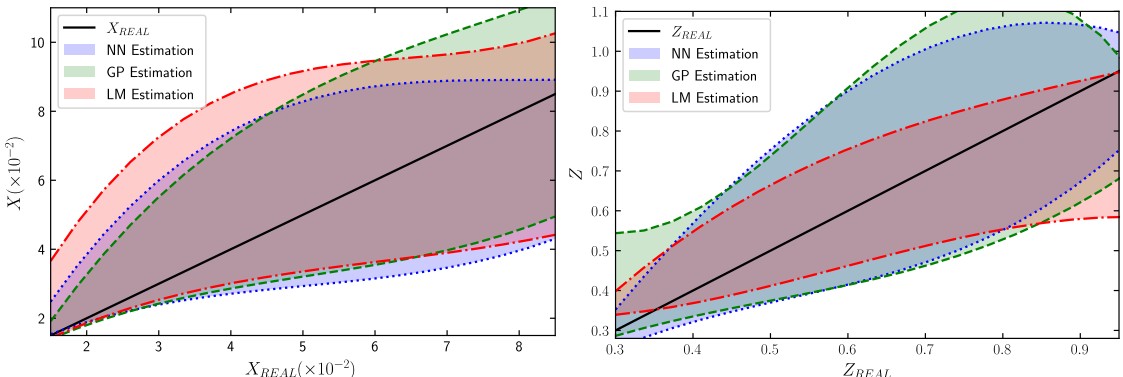

Figure 18: Correlation plots for $X = x_1$ (left) and $X = z$ (right). We defined the error bands following the strategy mentioned in App. A, and present the results for the neural network (NN, blue), Gaussian process (GP, green) and the linear method (LM, red) estimations.

## 5 Conclusions and outlook

In contrast with $e + p$ collisions, where measuring the final lepton grants direct access to the parton kinematics, in $p + p$ collisions there is no such a clear relation between the partonic and the hadronic momenta beyond the LO. In this work we have explored the reconstruction of the parton-level kinematics for the process $p + p \rightarrow \gamma + h$ using Machine-Learning (ML) tools, aiming to improve the current estimates of the underlying connection between the parton process and the measurable final state.

In first place, we implemented the calculation in a Monte-Carlo (MC) code with NLO QCD and LO QED accuracy. We relied on the FKS algorithm to cancel the infrared singularities, and the smooth cone isolation criteria to select those events with direct photons. This prescription is crucial to have access to cleaner information from the hard process.

Then, we studied different kinematical distributions with the purpose of identifying the regions with the largest number of events. After imposing selection cuts similar to those used by experimental collaborations, dynamical cuts were induced in the $x$ and $z$ distributions. These restrictions were taken into account when selecting events for analysing the correlations between experimentally-accessible quantities ($p_T$, $\eta$ and $\phi$ for the photon and pion) and the partonic momentum fraction. We realized that $x$ strongly depends on $p_T^\gamma$ (positive correlation) but not on the other variables, whilst $z$ exhibits a negative correlation with $p_T^\gamma$, a positive one with $p_T^\pi$ and a mild dependence with $\cos(\phi^\pi - \phi^\gamma)$.

After that, we applied ML algorithms to reconstruct the partonic variables $x_1$, $x_2$ and $z$. We started by introducing a proper discretization of the multi-differential cross-section w.r.t. the set of variables $\{p_T^\pi, p_T^\gamma, \eta^\pi, \eta^\gamma, \cos(\phi^\pi - \phi^\gamma)\}$, in order to have a reliable estimation of the higher-order corrections in each bin. For these distributions, we generated the data sets and explored three different ML reconstruction strategies: linear methods (LM), Gaussian Regression (GR) and Multi-Layer Perceptron (MLP). For the first two approaches, we introduced three bases of functions inspired by the results obtained from the analysis of two-dimensional correlations in Sec. 3.2. In all the cases, the reconstruction at LO QCD accuracy was very successful, and in agreement with the known analytical expressions. When dealing with the NLO QCD + LO QED corrections, the flexibility of the MLP approach leads to a very reliable reconstruction, achieving a better performance than the LM and comparable to the GR when using a sufficiently large basis. In particular, the LM results were highly-influenced by the abundance of data in the low-$x$ and low-$z$ region, leading to an unreliable fit when extrapolated outside

these regions.

The number of assumptions related to the setup of the MLP framework is rather limited, compared to the ones done for linear and Gaussian regression. In particular, we want to highlight that there was no need to introduce an specific basis of functions, which makes this approach fully process-independent and suitable for other analysis. Thus, this work can be regarded as a *proof-of-concept*, pointing towards a highly-automatized framework to include higher-order corrections in the reconstruction of the parton-level kinematics. For this reason, we prefer to center into the conceptual details and present an analysis based on a few ML approaches, rather than a deep study to find the optimal ML strategy to reach a fast and accurate reconstruction.

The reconstruction of the partonic momentum fractions shown in Fig. 16 can be used to ease the interpretation of the parton-level interactions. As we already mentioned, beyond LO, several processes contribute to a given observable. These processes contain a different number of particles in the final state, which are integrated over an extended phase-space. As a consequence, the naive LO interpretation of $x$ and $z$ is no longer valid when higher-order corrections enter into the game. Still, Eqs. (24)-(26) provide a probabilistic definition of an *equivalent momentum fraction*. The equivalent momentum fraction contains information about the differential cross-section: in fact, the different partonic processes, with different momentum fractions, are weighted by the cross-section. This can be used to simplify the treatment of Eq. (9), and consider a *LO-like approximation* in which the corrected PDFs/FFs are directly convoluted with the cross-section at a given *fixed* equivalent momentum fractions. As a result, this could lead to a more efficient computational implementation of the PDFs/FFs extraction, by-passing the complications of having several integrations/convolutions when dealing with higher-orders. We defer the explicit bench-marking of this implementation to future investigations [82].

In conclusion, the application of ML-inspired methods (and Neural Networks in particular) is suitable to unveil the partonic kinematics at hadron colliders, including also higher-order corrections. In this way, ML-assisted event reconstruction might allow to achieve a highly-precise description of the deepest constituents of matter and their interactions, complementing the current developments in other areas of theoretical particle physics.

## Acknowledgements

We would like to thank Germán Rodrigo, Andreas Schäfer, Leandro Cieri and Markus Diefenthaler for fruitful comments about the manuscript. Also, we thank Emma Torró and Miguel Villaplana for their illuminating comments about photon selection criteria at LHC.

**Author contributions**    All the authors equally contributed to the present manuscript.

**Funding information**    This research was partially supported by COST Action CA16201 (PARTICLEFACE) and MCIN/AEI/10.13039/501100011033, Grant No. PID2020-114473GB-I00. The work of D. F. R.-E. and R. J. H.-P. is supported by CONACyT (México) through the Project No. A1- S-33202 (Ciencia Básica), No. 320856 (Paradigmas y Controversias de la Ciencia 2022) and Ciencia de Frontera 2021-2042; in addition by PROFAPI 2022 Grant No. PRO_A1_024 (Universidad Autónoma de Sinaloa). Besides, R. J. H.-P. is also funded by Sistema Nacional de Investigadores from CONACyT. P.Z. acknowledges support from the Deutsche Forschungsgemeinschaft (DFG, German Research Foundation) - Research Unit FOR 2926, grant number 430915485. The work of G.S. was partially supported by Universidad de Salamanca through Programas Propios II, by Ministerio de Ciencia e Innovación under Contract No.

PID2019-105439GB-C22/AEI/10.13039/501100011033 and by EU Horizon 2020 research and innovation program, STRONG-2020 project, under grant agreement No. 824093.

# A    Details about the error propagation strategies

The theoretical predictions obtained in the context of perturbation theory have an explicit dependence on the renormalization and factorization scales. This dependence is often used to quantify the uncertainties of the prediction, due to missing higher-order corrections. On top of that, the uncertainties associated to the PDFs/FFs fitting should also be considered in the calculation of the theoretical cross-section. The full propagation of these errors would require to run the Monte-Carlo integration for several hundreds of replicas and then perform the associated statistical analysis, with a significant computational cost.

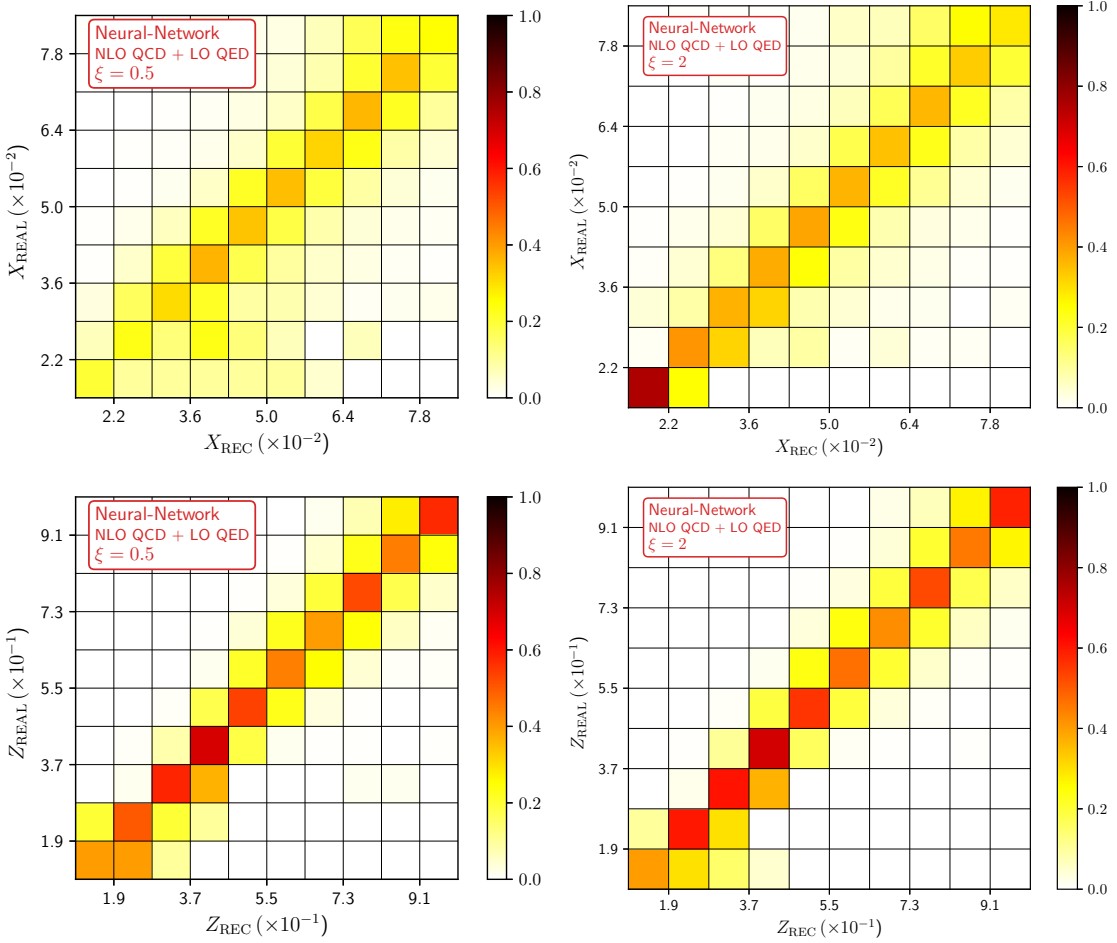

Figure 19: First row: Comparison of the momentum fractions $X_{\text{REAL}}$ vs. $X_{\text{REC}}$ obtained with MLP neural networks (using the parameters given in Table 2) for different renormalization/factorization scales, $\mu_R = \mu_F \equiv \xi\mu$. We set $\xi = 1/2$ ($\xi = 2$) in the left (right) plot. Second row: same as in the first row, but for $Z_{\text{REAL}}$ vs. $Z_{\text{REC}}$.

In our case, the scale uncertainty at the level of one-dimensional distributions (as the ones described in Sec. 3.1) is dominant w.r.t. the error induced by variations in the PDFs/FFs. In fact, the scale dependence accounts for $\mathcal{O}(30-50\%)$ corrections [25] whilst changing the PDFs/FFs could induce $\mathcal{O}(10\%)$ error. Furthermore, the analysis presented in Ref. [38]

refers to using *different* PDF/FF sets, which can be considered as an extreme or worst-case scenario for estimating PDF/FF fit uncertainties. If we keep the same PDF/FF, this error will be drastically reduced due to the high precision of the current fits. For all these reasons, in this work we only propagate the errors associated to the scale variation.

The propagation of the error due to the scale uncertainty is achieved by running the Monte-Carlo simulation for three different scenarios. There is no unique way to estimate the error of the reconstruction, thus we propose two strategies. The starting point in both methods consists in fixing the condition $\mu_R = \mu_I = \mu_F \equiv \xi\mu$ and generating the datasets

$$\mathcal{D}^{(\xi)} = \{\bar{p}_T^\gamma, \bar{p}_T^\pi, \bar{\eta}^\gamma, \bar{\eta}^\pi, \overline{\cos}(\phi^\pi - \phi^\gamma), x_1, x_2, z, \sigma(\xi\mu)\}, \tag{44}$$

following the binning procedure described in Sec. 4.1. As it is conventionally done, we consider $\xi \in \{1/2, 1, 2\}$. In Eq. (44), the cross-section per bin ($\sigma$) depends on the renormalization/factorization scales and this induces an implicit dependence on $\{x_1, x_2, z\}$ according to Eqs. (24)-(26).

Our first method puts more emphasis on the quality of the correlation among real and reconstructed variables. We consider the target functions trained with the dataset $\mathcal{D}^{(\xi=1)}$, i.e. $X_{\text{REC}} \equiv X_{\text{REC}}^{(\xi=1)}$, but create three different correlations plots. For instance, given

$$(p_j, x_1, x_2, z, \sigma) \in \mathcal{D}^{(\xi)}, \tag{45}$$

we compare $x_1$ ($y$-axis) w.r.t. $X_{1,\text{REC}}(p_j)$ ($x$-axis). When we use the dataset with $\xi = 1$, the plots shown in Figs. 12-16 are recovered. In these plots, the datapoints tend to group in the diagonal because the training optimizes the reconstruction. However, the correlation plots obtained with the datasets $\mathcal{D}^{(\xi=1/2)}$ and $\mathcal{D}^{(\xi=2)}$ use a reconstruction function that is not optimized for those datapoints. As a consequence, a deviation from the diagonal is expected. Hence, we can estimate the effect of scale uncertainty by comparing the three correlation plots. It is worth mentioning that this approach mainly provides a qualitative estimation of the reconstruction error, since a clearly defined diagonal would indicate a nearly perfect reconstruction.

For the sake of simplicity, we applied this procedure for describing the error propagation in the case of using MLP neural network algorithm. We show in Fig. 19 a comparison of the correlation plots $X_{\text{REC}}$ vs $X_{\text{REAL}}$ ($Z_{\text{REC}}$ vs $Z_{\text{REAL}}$) in the first (second) row, using $\xi = 1/2$ (left) and $\xi = 2$ (right) respectively. On one side, for $X_{\text{REC}}$ vs $X_{\text{REAL}}$, the correlation gets diluted and more extra-diagonal bins are populated. This suggests that the reconstruction of $X = x_1$ is very sensitive and enhances the scale uncertainties. On the other side, $Z$ seems to be more stable and rather independent on the scale variations because both plots present a clear diagonal.

A variation of this strategy consists in comparing the one-dimensional distributions in $X_{\text{REC}}$ for the different datasets. Since it turns out that this methodology provides a clearer error estimation, we present the results in the main text (Sec. 4.5). Explicitly, we train the reconstructed momentum fractions using the dataset $\mathcal{D}^{(\xi=1)}$. Then, we take the values of the binned datasets $\mathcal{D}^{(\xi)}$ and replace the real momentum fractions by the reconstructed ones, i.e.

$$(p_j, x_1, x_2, z, \sigma) \rightarrow (p_j, X_{1,\text{REC}}(p_j), X_{2,\text{REC}}(p_j), Z_{\text{REC}}(p_j), \sigma). \tag{46}$$

Using these datapoints, we draw the histograms for $\xi = \{1/2, 1, 2\}$ and compare the results. In this way, we are able to provide an error band that quantifies the impact of the scale variation in the $X$- and $Z$-spectra.

The second error estimation strategy consists in using the datasets $\mathcal{D}^{(\xi)}$ to train different target functions;

$$X_{1,\text{REC}}^{(\xi)} \quad := \quad \bar{\mathcal{V}}_{\text{Exp}}^{(\xi)} \longrightarrow \bar{X}_{1,REAL}^{(\xi)} = \{(x_1)_j\} \subset \mathcal{D}^{(\xi)}, \tag{47}$$

$$X_{2,\text{REC}}^{(\xi)} \quad := \quad \bar{\mathcal{V}}_{\text{Exp}}^{(\xi)} \longrightarrow \bar{X}_{2,REAL}^{(\xi)} = \{(x_2)_j\} \subset \mathcal{D}^{(\xi)}, \tag{48}$$

$$Z_{\text{REC}}^{(\xi)} \quad := \quad \bar{\mathcal{V}}_{\text{Exp}}^{(\xi)} \longrightarrow \bar{Z}_{REAL}^{(\xi)} = \{(z)_j\} \subset \mathcal{D}^{(\xi)}. \tag{49}$$

This procedure leads to three functions for reconstructing each momentum fraction: given a kinematic point in the grid, $p_j \in \bar{\mathcal{V}}_{\text{Exp}}$, we have

$$X(p_j) \equiv \{X_{\text{REC}}^{(\xi=2)}(p_j), X_{\text{REC}}^{(\xi=1)}(p_j), X_{\text{REC}}^{(\xi=1/2)}(p_j)\}, \tag{50}$$

and define

$$X_{\text{REC}}(p_j) = \overline{X(p_j)} \pm \frac{\max(X(p_j)) - \min(X(p_j))}{2} \equiv \overline{X(p_j)} \pm \Delta X(p_j), \tag{51}$$

with $X \in \{X_1, X_2, Z\}$ and $\overline{X(p_j)}$ the mean value of the list $X(p_j)$. This approach mainly estimates the error of the fitting procedure due to the scale uncertainties in the cross-section and the subsequent definition of the weighted momentum fractions in Eqs. (24)-(26). Again, we present the results for the different reconstruction methods in Sec. 4.5.

## B  Coefficients for the Linear Method

For completeness, we present the coefficients associated to the linear regression for each of the three bases studied in Subsec. 4.2. We restrict our attention to the fit of the data sets at NLO QCD + LO QED accuracy, since the LO contributions were perfectly in agreement with the analytical LO formulae. In Tab. 4 we present the coefficients of the most general basis, Eq. (38), that reproduce the plots in the upper row of Fig. 13. The parameters of the *physically-motivated* basis, given by Eq. (38) with the constraints of Eqs. (39)-(40), are in Tabs. 5 and 7 for $x$ and $z$, respectively. The corresponding correlation with the real MC variables can be seen in the middle row of Fig. 13. Finally, the coefficients for the *LO-inspired* basis, associated to the constraints in Eqs. (41)-(42), can be found in Tab. 6. These fall short in the quality of the fit, as we can appreciate from the lower row of Fig. 13.

## C  Comparison of different NN architectures

We summarize here some results that were obtained before the *optimal* architecture described in Subsec. 4.4 was found. In Tab. 3 we present the parameters corresponding to three different tests implemented.

Table 3: Architectures for the MLP of three different tests for the reconstruction of the momentum fractions at NLO in QCD. All parameters are taken to be the same for $X_{\text{REC}}$ and $Z_{\text{REC}}$.

| Parameters | TEST 1 | TEST 2 | TEST 3 |
|---|---|---|---|
| # hidden layers | 2 | 4 | 3 |
| # neurons/layer | 50 | 100 | 100 |
| tolerance | $10^{-2}$ | $10^{-2}$ | $10^{-3}$ |
| max. number of iterations | $10^8$ | $10^8$ | $10^9$ |
| # iterations w/o change | 14,000 | 21,000 | 100,000 |

In TEST1 (upper row of Fig. 20), we use a lower number of neurons/layer and less layers than for obtaining the results in Fig. 16. We find a poor agreement between the real and reconstructed quantities, in particular for low-$z$ bins. An improvement is achieved by increasing the number of layers and neurons/layer (TEST2), while simultaneously requiring the NN to

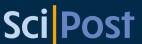

Figure 20: Comparison of the momentum fractions $X_{\text{REAL}}$ vs. $X_{\text{REC}}$ (left) and $Z_{\text{REAL}}$ vs. $Z_{\text{REC}}$ (right) obtained with MLP at NLO QCD + LO QED accuracy. The parameters for TEST1 (upper row), TEST2 (middle row) and TEST3 (lower row) are given in Table 3.

Table 4: Coefficients for the LM with the general basis expressed in Eq. (38) for both $x$ and $z$ momentum fractions.

| Coefficient | $X_{REC}$ (NLO) | $Z_{REC}$ (NLO) | Coefficient | $X_{REC}$ (NLO) | $Z_{REC}$ (NLO) |
|---|---|---|---|---|---|
| $a_1^Y$ | $-5.7 \times 10^1$ | $-1.1 \times 10^3$ | $c_{68}^Y$ | $-6.7 \times 10^{-4}$ | $7.3 \times 10^{-2}$ |
| $a_2^Y$ | $7.2 \times 10^1$ | $3.3 \times 10^2$ | $c_{69}^Y$ | $3.6 \times 10^{-3}$ | $6.6 \times 10^{-2}$ |
| $a_3^Y$ | $5.4 \times 10^0$ | $-5.6 \times 10^1$ | $c_{77}^Y$ | $-1.2 \times 10^{-4}$ | $-1.9 \times 10^{-4}$ |
| $a_4^Y$ | $-2.4 \times 10^0$ | $2.7 \times 10^0$ | $c_{78}^Y$ | $-7.9 \times 10^{-3}$ | $3.6 \times 10^{-2}$ |
| $a_5^Y$ | $-4.9 \times 10^0$ | $8.0 \times 10^1$ | $c_{79}^Y$ | $-5.6 \times 10^{-3}$ | $5.5 \times 10^{-3}$ |
| $a_6^Y$ | $-2.8 \times 10^{-2}$ | $-1.2 \times 10^{-1}$ | $c_{88}^Y$ | $-1.3 \times 10^0$ | $1.1 \times 10^1$ |
| $a_7^Y$ | $3.8 \times 10^{-2}$ | $1.6 \times 10^{-2}$ | $c_{89}^Y$ | $5.6 \times 10^{-1}$ | $3.8 \times 10^{-2}$ |
| $a_8^Y$ | $5.2 \times 10^0$ | $-5.6 \times 10^1$ | $c_{99}^Y$ | $1.9 \times 10^{-1}$ | $-2.5 \times 10^0$ |
| $a_9^Y$ | $-2.1 \times 10^0$ | $9.4 \times 10^{-1}$ | $d_{11}^Y$ | $-6.9 \times 10^2$ | $4.4 \times 10^3$ |
| $b_1^Y$ | $-6.8 \times 10^1$ | $-1.2 \times 10^3$ | $d_{12}^Y$ | $2.5 \times 10^3$ | $-1.3 \times 10^4$ |
| $b_2^Y$ | $5.8 \times 10^1$ | $5.2 \times 10^2$ | $d_{13}^Y$ | $1.9 \times 10^0$ | $2.3 \times 10^2$ |
| $b_3^Y$ | $4.9 \times 10^0$ | $-5.6 \times 10^1$ | $d_{14}^Y$ | $6.3 \times 10^0$ | $3.8 \times 10^2$ |
| $b_4^Y$ | $-2.2 \times 10^0$ | $-1.6 \times 10^{-1}$ | $d_{17}^Y$ | $3.2 \times 10^{-1}$ | $1.7 \times 10^0$ |
| $b_6^Y$ | $-3.1 \times 10^{-2}$ | $-9.1 \times 10^{-2}$ | $d_{18}^Y$ | $4.0 \times 10^{-1}$ | $2.6 \times 10^2$ |
| $b_7^Y$ | $3.5 \times 10^{-2}$ | $3.2 \times 10^{-2}$ | $d_{19}^Y$ | $9.7 \times 10^0$ | $3.4 \times 10^2$ |
| $b_8^Y$ | $4.7 \times 10^0$ | $-5.7 \times 10^1$ | $d_{22}^Y$ | $-7.6 \times 10^2$ | $-3.2 \times 10^3$ |
| $b_9^Y$ | $-1.9 \times 10^0$ | $-2.2 \times 10^0$ | $d_{23}^Y$ | $-3.2 \times 10^1$ | $1.4 \times 10^2$ |
| $c_{11}^Y$ | $-4.9 \times 10^2$ | $2.4 \times 10^3$ | $d_{24}^Y$ | $-1.4 \times 10^1$ | $1.3 \times 10^1$ |
| $c_{12}^Y$ | $1.9 \times 10^3$ | $-9.8 \times 10^3$ | $d_{26}^Y$ | $6.5 \times 10^{-1}$ | $-5.5 \times 10^0$ |
| $c_{13}^Y$ | $1.7 \times 10^0$ | $2.4 \times 10^2$ | $d_{28}^Y$ | $-2.5 \times 10^1$ | $1.5 \times 10^2$ |
| $c_{14}^Y$ | $6.2 \times 10^0$ | $3.6 \times 10^2$ | $d_{29}^Y$ | $-1.1 \times 10^1$ | $7.9 \times 10^0$ |
| $c_{17}^Y$ | $1.8 \times 10^{-1}$ | $1.6 \times 10^0$ | $d_{33}^Y$ | $-1.1 \times 10^0$ | $1.1 \times 10^1$ |
| $c_{18}^Y$ | $1.4 \times 10^{-1}$ | $2.6 \times 10^2$ | $d_{34}^Y$ | $4.1 \times 10^{-1}$ | $-1.3 \times 10^0$ |
| $c_{19}^Y$ | $9.9 \times 10^0$ | $3.1 \times 10^2$ | $d_{36}^Y$ | $-6.0 \times 10^{-4}$ | $6.7 \times 10^{-2}$ |
| $c_{22}^Y$ | $-7.2 \times 10^2$ | $-3.0 \times 10^3$ | $d_{37}^Y$ | $-9.6 \times 10^{-3}$ | $3.4 \times 10^{-2}$ |
| $c_{23}^Y$ | $-3.1 \times 10^1$ | $1.5 \times 10^2$ | $d_{39}^Y$ | $5.4 \times 10^{-1}$ | $-3.6 \times 10^{-1}$ |
| $c_{24}^Y$ | $-1.4 \times 10^1$ | $2.5 \times 10^1$ | $d_{44}^Y$ | $4.7 \times 10^{-1}$ | $-2.4 \times 10^0$ |
| $c_{26}^Y$ | $5.3 \times 10^{-1}$ | $-4.3 \times 10^0$ | $d_{46}^Y$ | $2.8 \times 10^{-3}$ | $8.1 \times 10^{-2}$ |
| $c_{28}^Y$ | $-2.5 \times 10^1$ | $1.5 \times 10^2$ | $d_{47}^Y$ | $-6.1 \times 10^{-3}$ | $3.0 \times 10^{-3}$ |
| $c_{29}^Y$ | $-1.1 \times 10^1$ | $1.9 \times 10^1$ | $d_{48}^Y$ | $4.9 \times 10^{-1}$ | $-1.3 \times 10^0$ |
| $c_{33}^Y$ | $-1.2 \times 10^0$ | $1.0 \times 10^1$ | $d_{66}^Y$ | $1.5 \times 10^{-5}$ | $-2.1 \times 10^{-4}$ |
| $c_{34}^Y$ | $3.6 \times 10^{-1}$ | $-7.7 \times 10^{-1}$ | $d_{67}^Y$ | $1.9 \times 10^{-4}$ | $-1.0 \times 10^{-3}$ |
| $c_{36}^Y$ | $-6.6 \times 10^{-4}$ | $6.9 \times 10^{-2}$ | $d_{68}^Y$ | $-6.1 \times 10^{-4}$ | $7.2 \times 10^{-2}$ |
| $c_{37}^Y$ | $-9.4 \times 10^{-3}$ | $3.5 \times 10^{-2}$ | $d_{69}^Y$ | $3.6 \times 10^{-3}$ | $7.0 \times 10^{-2}$ |
| $c_{39}^Y$ | $4.8 \times 10^{-1}$ | $5.3 \times 10^{-2}$ | $d_{77}^Y$ | $-1.3 \times 10^{-4}$ | $-1.3 \times 10^{-4}$ |
| $c_{44}^Y$ | $5.6 \times 10^{-1}$ | $-3.3 \times 10^0$ | $d_{78}^Y$ | $-8.1 \times 10^{-3}$ | $3.6 \times 10^{-2}$ |
| $c_{46}^Y$ | $2.8 \times 10^{-3}$ | $7.6 \times 10^{-2}$ | $d_{79}^Y$ | $-5.5 \times 10^{-3}$ | $1.2 \times 10^{-3}$ |
| $c_{47}^Y$ | $-6.3 \times 10^{-3}$ | $7.8 \times 10^{-3}$ | $d_{88}^Y$ | $-1.2 \times 10^0$ | $1.1 \times 10^1$ |
| $c_{48}^Y$ | $4.4 \times 10^{-1}$ | $-8.0 \times 10^{-1}$ | $d_{89}^Y$ | $6.1 \times 10^{-1}$ | $-3.8 \times 10^{-1}$ |
| $c_{66}^Y$ | $2.2 \times 10^{-5}$ | $-2.2 \times 10^{-4}$ | $d_{99}^Y$ | $1.0 \times 10^{-1}$ | $-1.6 \times 10^0$ |
| $c_{67}^Y$ | $1.4 \times 10^{-4}$ | $-7.9 \times 10^{-4}$ | | | |

see no variation of the cost function (within a given tolerance) through a larger number of iterations. As seen in Fig. 20 (middle row), this gives a better reconstruction, thought it is still far from ideal. A third example, TEST3, reinforces the conditions for convergence and returns a significantly improved result (lower row of Fig. 20). Each step towards a more complex

architecture and more stringent requirements for convergence is translated into an increase of the computational time required for the training. These, and other trials, have guided us to the selection of the *best* architecture for our task, summarised in Tab. 2.

Table 5: Coefficients for the LM with the *physically-motivated* basis expressed in Eq. (38) with the constraints given in Eq. (39) and Eq. (40) for the $x$ momentum fraction.

| Coefficient | $X_{REC}$ (NLO) | Coefficient | $X_{REC}$ (NLO) |
|---|---|---|---|
| $a_1^Y$ | $5.5 \times 10^1$ | $c_{48}^Y$ | $4.2 \times 10^{-1}$ |
| $a_2^Y$ | $1.4 \times 10^2$ | $c_{77}^Y$ | $-1.0 \times 10^{-4}$ |
| $a_3^Y$ | $5.4 \times 10^0$ | $c_{78}^Y$ | $-8.0 \times 10^{-3}$ |
| $a_4^Y$ | $-2.3 \times 10^0$ | $c_{79}^Y$ | $-5.4 \times 10^{-3}$ |
| $a_5^Y$ | $-8.4 \times 10^0$ | $c_{88}^Y$ | $-1.3 \times 10^0$ |
| $a_7^Y$ | $5.6 \times 10^{-2}$ | $c_{89}^Y$ | $5.3 \times 10^{-1}$ |
| $a_8^Y$ | $5.2 \times 10^0$ | $c_{99}^Y$ | $2.5 \times 10^{-1}$ |
| $a_9^Y$ | $-1.8 \times 10^0$ | $d_{11}^Y$ | $-4.1 \times 10^2$ |
| $b_1^Y$ | $6.3 \times 10^1$ | $d_{12}^Y$ | $-6.4 \times 10^2$ |
| $b_2^Y$ | $1.4 \times 10^2$ | $d_{13}^Y$ | $3.9 \times 10^0$ |
| $b_3^Y$ | $4.9 \times 10^0$ | $d_{14}^Y$ | $-7.4 \times 10^0$ |
| $b_4^Y$ | $-2.1 \times 10^0$ | $d_{17}^Y$ | $-5.6 \times 10^{-1}$ |
| $b_7^Y$ | $5.8 \times 10^{-2}$ | $d_{18}^Y$ | $2.5 \times 10^0$ |
| $b_8^Y$ | $4.7 \times 10^0$ | $d_{19}^Y$ | $-8.0 \times 10^0$ |
| $b_9^Y$ | $-1.6 \times 10^0$ | $d_{22}^Y$ | $-6.5 \times 10^2$ |
| $c_{11}^Y$ | $-3.2 \times 10^2$ | $d_{23}^Y$ | $-3.2 \times 10^1$ |
| $c_{12}^Y$ | $-6.0 \times 10^2$ | $d_{24}^Y$ | $-1.4 \times 10^1$ |
| $c_{13}^Y$ | $4.1 \times 10^0$ | $d_{28}^Y$ | $-2.5 \times 10^1$ |
| $c_{14}^Y$ | $-7.3 \times 10^0$ | $d_{29}^Y$ | $-1.0 \times 10^1$ |
| $c_{17}^Y$ | $-4.8 \times 10^{-1}$ | $d_{33}^Y$ | $-1.1 \times 10^0$ |
| $c_{18}^Y$ | $2.6 \times 10^0$ | $d_{34}^Y$ | $3.8 \times 10^{-1}$ |
| $c_{19}^Y$ | $-7.8 \times 10^0$ | $d_{37}^Y$ | $-9.6 \times 10^{-3}$ |
| $c_{47}^Y$ | $-6.1 \times 10^{-3}$ | | |

Table 6: Coefficients for the LM with the *LO-inspired* basis expressed in Eqs. (41) and (42) for both $x$ and $z$ momentum fractions.

| Coefficient | $X_{REC}$ (NLO) | Coefficient | $Z_{REC}$ (NLO) |
|---|---|---|---|
| $c_{13}^Y$ | $3.8 \times 10^0$ | $c_{26}^Y$ | $-2.5 \times 10^{-1}$ |
| $c_{14}^Y$ | $4.7 \times 10^{-1}$ | $d_{26}^Y$ | $-5.2 \times 10^{-2}$ |
| $c_{23}^Y$ | $2.0 \times 10^{-1}$ | $b_6^Y$ | $2.0 \times 10^{-3}$ |
| $c_{24}^Y$ | $1.6 \times 10^0$ | $b_2^Y$ | $5.4 \times 10^0$ |
| $d_{13}^Y$ | $3.6 \times 10^0$ | | |
| $d_{14}^Y$ | $1.7 \times 10^{-1}$ | | |
| $d_{23}^Y$ | $-5.4 \times 10^{-1}$ | | |
| $d_{24}^Y$ | $9.1 \times 10^{-1}$ | | |

Table 7: Same as Tab. 5, now for the $z$ momentum fraction.

| Coefficient | $Z_{REC}$ (NLO) | Coefficient | $Z_{REC}$ (NLO) |
|---|---|---|---|
| $a_2^Y$ | $-2.0 \times 10^2$ | $c_{67}^Y$ | $-1.7 \times 10^2$ |
| $a_3^Y$ | $-4.1 \times 10^1$ | $c_{68}^Y$ | $-4.2 \times 10^1$ |
| $a_4^Y$ | $1.5 \times 10^1$ | $c_{69}^Y$ | $1.3 \times 10^1$ |
| $a_5^Y$ | $5.0 \times 10^1$ | $c_{88}^Y$ | $-1.7 \times 10^{-2}$ |
| $a_6^Y$ | $-3.5 \times 10^{-2}$ | $c_{89}^Y$ | $-4.2 \times 10^1$ |
| $a_8^Y$ | $-4.1 \times 10^1$ | $c_{99}^Y$ | $9.1 \times 10^0$ |
| $a_9^Y$ | $1.2 \times 10^1$ | $d_{11}^Y$ | $6.7 \times 10^2$ |
| $b_2^Y$ | $6.6 \times 10^2$ | $d_{22}^Y$ | $1.2 \times 10^3$ |
| $b_3^Y$ | $1.4 \times 10^3$ | $d_{23}^Y$ | $4.0 \times 10^1$ |
| $b_4^Y$ | $4.4 \times 10^1$ | $d_{24}^Y$ | $1.0 \times 10^1$ |
| $b_6^Y$ | $7.2 \times 10^0$ | $d_{26}^Y$ | $3.2 \times 10^{-2}$ |
| $b_8^Y$ | $7.0 \times 10^{-2}$ | $d_{28}^Y$ | $4.0 \times 10^1$ |
| $b_9^Y$ | $4.5 \times 10^1$ | $d_{29}^Y$ | $9.6 \times 10^0$ |
| $c_{11}^Y$ | $6.8 \times 10^0$ | $d_{33}^Y$ | $1.0 \times 10^1$ |
| $c_{22}^Y$ | $9.6 \times 10^0$ | $d_{34}^Y$ | $-1.5 \times 10^0$ |
| $c_{23}^Y$ | $-1.0 \times 10^0$ | $d_{36}^Y$ | $2.0 \times 10^{-2}$ |
| $c_{24}^Y$ | $2.1 \times 10^{-2}$ | $d_{39}^Y$ | $-6.0 \times 10^{-1}$ |
| $c_{26}^Y$ | $-1.9 \times 10^{-1}$ | $d_{44}^Y$ | $-2.8 \times 10^0$ |
| $c_{28}^Y$ | $-3.6 \times 10^0$ | $d_{46}^Y$ | $6.7 \times 10^{-3}$ |
| $c_{29}^Y$ | $6.8 \times 10^{-3}$ | $d_{48}^Y$ | $-1.8 \times 10^0$ |
| $c_{33}^Y$ | $-1.2 \times 10^0$ | $d_{66}^Y$ | $-1.4 \times 10^{-4}$ |
| $c_{34}^Y$ | $-9.7 \times 10^{-5}$ | $d_{67}^Y$ | $-1.2 \times 10^{-4}$ |
| $c_{36}^Y$ | $-8.5 \times 10^{-5}$ | $d_{68}^Y$ | $2.0 \times 10^{-2}$ |
| $c_{39}^Y$ | $2.1 \times 10^{-2}$ | $d_{69}^Y$ | $4.9 \times 10^{-3}$ |
| $c_{44}^Y$ | $5.1 \times 10^{-3}$ | $d_{88}^Y$ | $1.0 \times 10^1$ |
| $c_{46}^Y$ | $9.7 \times 10^0$ | $d_{89}^Y$ | $-8.4 \times 10^{-1}$ |
| $c_{48}^Y$ | $-3.9 \times 10^{-1}$ | $d_{99}^Y$ | $-1.7 \times 10^0$ |
| $c_{66}^Y$ | $-2.6 \times 10^0$ | | |

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
