# Peer review of "Reconstructing partonic kinematics at colliders with Machine Learning"

_SciPost Physics, doi:SciPost Phys. Core 5, 049 (2022)_

## Round 1 · Referee Report · Anonymous (Referee 1) · 2022-3-18

Strengths

  1. Problems that authors are going to solve are clearly stated.
  2. Simple machine learning technique is used to solve the problems.

Weaknesses

  1. Uncertainties on their parton kinematics predictions are not discussed.
  2. The selected kinematic region of the process may not be visible at the LHC.
  3. The machine learning solution of their problems may have a numerical solution.

Report

This paper is a proof-of-concept study of reconstructing parton-level kinematics in proton-proton collisions only using measurable final state data.
Authors attempt to reconstruct the initial state and final state momentum fractions of a hard process in a proton+proton->pion+photon (pp->pi+gamma) process as an illustrative example.
For this purpose, the authors introduced supervised regression techniques based on particle-level Monte-Carlo event simulation and three regression models: linear regression, Gaussian regression, and neural network.
Their method could reconstruct the parton-level kinematics with reasonable accuracy.

While this work is an interesting application of machine learning to collider physics, I think the following points must be addressed before I recommend this work for publication.

Requested changes

  1. The selected process (pp->pi+gamma) with the mild selection criterion in Eq. 13 may be highly contaminated by busy hadron activities and pile-ups in (high-luminosity) LHC. Also, the trigger systems of ATLAS and CMS may wash out some of the kinematic regions considered in this paper. For example, this ATLAS report (arXiv:1909.00761) and CMS report (http://cds.cern.ch/record/2668901) say that the trigger efficiencies of photons are not good for pT < 15 GeV. Delphes uses 10 GeV cuts to photon for both ATLAS and CMS detector simulations. Do you have some idea about those?

  2. Renormalization and factorization scale uncertainties and PDF/FF fitting uncertainty are not discussed. The parton level kinematics estimators are sensitive to those functions since it is trained by supervised learning from the dataset using those functions. The systematic uncertainty propagation should be reported together to strengthen the author's claim about the reconstructibility of the parton level kinematics.

  3. Please state the loss functions for training the models.

  4. If authors use mean square errors (MSE) for fitting x and z in their training dataset, I think authors could give us an asymptotic solution of their regressor. The regression using MSE loss has an asymptotic solution: the expectation of x (or z) given inputs of neural networks. Since the authors showed closed forms of differential cross-sections of their process and its Monte-Carlo simulation, I think the computation of the asymptotic solution is relatively straightforward.

  5. The lower right of Figure 16 shows that some components still fail to be reconstructed. Have you tried larger-sized networks and used more training samples? There might be some underfitting since fitting performance keeps improving when you enlarge the network size.

  6. Which dataset is used for drawing the correlation plots: fig 13, fig 14, fig 15, and fig 16? If the same simulated dataset is used, I think getting diagonal lines on those plots are somewhat natural. However, the simulation is always limited and different from nature, so sizable bias may occur in reality. Do you have any estimation of such bias by applying your trained networks to other simulated datasets from other Monte Carlo event simulations, such as Herwig, Pythia, Sherpa, and so on?

Minor comments: - Page 15: MLP is a model of function, not an algorithm.

  • Section 3.2 says that the following analyses are based on RHIC kinematics. If analysis on section 4 also uses data simulated at the RHIC energy scale, I think it's better to show that in the plots to improve readability because several energy scales are considered in previous sections.

  • Section 4.5 is about future works. The analysis is not done in this paper yet, so it's better to move it to the conclusions and outlook section.

---

## Round 1 · Referee Report · Anonymous (Referee 2) · 2022-3-27

Report

This paper uses machine learning to reconstruct the parton kinematics in proton proton collisions that produce a hadron and a photon. It is similar in spirit to the ep studies in [3,4], only there are not classical methods to compare with (as the kinematics of the initial state are less constrained by the final state). Overall, I found the paper to contain a lot of useful information, but many of the descriptions are not concise. For example, I'm not sure how much of Sec. 2 and 3 is really necesary to have in the main body of the paper. I also don't think it is necesary to explain what a neural network is in the main body.

  • The references in the first paragraph are a bit random. This is not a problem per se, but I would encourage the authors to reconsider their list. I found the mention about EIC to also be a bit strange. It is true that there is interest in the EIC community to use AI/ML methods, but this is a relatively small part of the larger community (and the one referenced workshop is balanced by countless workshops in the context of the LHC).

  • Please use vectorized graphics.

  • I was missing a discussion / demonstration of model dependence. If you train with one model and test with another, how universal are the results?

  • Related: the results seem very process dependent. Would one need to retrain for every process? What about processes that are ambiguous?

  • It was not completely obvious to me what this method would be used for. In e+e- and ep, the event kinematics can be reconstructed and are used as inputs to theory comparisons / extractions. Would you please say a bit more about this? (sorry if it is alreayd included and I missed it!)

---

## Round 2 · Referee Report · Anonymous (Referee 1) · 2022-6-7

Report

Thanks for considering the issues that I've raised and dealing with those. I think most of the problems are answered, and the draft is almost ready for publication. But I have one minor comment:

Requested changes

A3. Section 4.4 does not contain the details on the loss functions. Maybe the authors uploaded the wrong version of the draft?

Additional replies:

A4. Thanks for the clarification. Since the closed form of the analytic solution is not available and direct evaluation of the asymptotic solution is less trivial, this method can be regarded as a numerical solution to the problem.

A6. OK. I agree that the plots are sufficient for validating your model conditioned on the simulation. It reconstructs the kinematics with reasonable accuracy when the simulation exactly depicts the physics behind the test dataset. In the LO case, there is no ambiguity, so MLP perfectly reconstructs the kinematics. But in the NLO case, the ambiguity in NLO kinematics smears the correlation plot as you said.
But this kind of supervised regression is always conditioned on the simulation; I just quickly had a feeling that those plots were somewhat optimistic since the simulation bias can degrade the fitting performance at any point.

  • validity: -
  • significance: -
  • originality: -
  • clarity: -
  • formatting: -
  • grammar: -

Author:  German Sborlini  on 2022-06-09  [id 2570]

(in reply to Report 1 on 2022-06-07)

Dear Referee,

Thank you very much for reviewing again our paper.

Regarding your question, in Sec. 4.4 we briefly mentioned the idea behind the minimization strategy used by scikit-learn.

More in details, the package allows us to deal with the score function, which should be maximized to identify the optimal fit (thus, it is related to the inverse of the cost function). The default score function used in the MLP approach is the Pearson correlation coefficient.

In any case, the physical conclusions of this paper are not dependent on the details of the minimization process neither the explicit score/cost function. An analysis of this topic goes far beyond the scope of the paper, and would enter more into the field of computer science.

---

## Round 2 · Referee Report · Anonymous (Referee 2) · 2022-6-24

Report

Thank you for taking into account my feedback. It seems I was not completely clear, so I will followup here. It would be very helpful to me if you would please respond point-by-point; this will make it easier to check that you have implemented all of my feedback (it seems like some points were missed). Hopefully resolving this round of comments will be fast and I will be able to recommend publication soon.

  • Overall, I found the paper to contain a lot of useful information, but many of the descriptions are not concise. For example, I'm not sure how much of Sec. 2 and 3 is really necessary to have in the main body of the paper. Please consider moving some of this to an appendix.

  • Additionally: It is necessary to explain what a neural network is in the main body.

  • The references in the first paragraph seem more random than before unfortunately. I don't really see why you need to reference websites - footnote 1 is really strange. At this point, I won't insist further, but please at least have a look again.

  • Please use vectorized graphics.

  • Let me clarify what I mean by model dependence. You use a particular PDF and FF for the training dataset. How dependent are you on varying the training / testing dataset? This seems to be at least partially addressed in Appendix A and given that it is central to the paper, I don't understand why it is merely an appendix.

  • "A comment comparing the kinematics of e+e-/ep versus pp collisions is included in the main text." -> great, thank you! Would you please point me to it so I don't have to dig in this long paper to see what you wrote?

---

## Round 2 · Author Response

Dear Editor,

First of all, we would like to thank the referees for their reports and their important suggestions. We seriously considered all the mentioned issues and modified our manuscript accordingly. In particular, we implemented the following minor changes:

  1. We indicated the energy and kinematics used for generating the events in Figs. 6-11 in Sec. 3.2.
  2. We moved the discussion of Sec 4.5 to the conclusions.
  3. We corrected some typos present in the text.
  4. We modified some references in the Introduction, and clarified certain phrases.

Regarding the major changes requested by the referees, we present a full and detailed list in the "List of changes" section. We have addressed all the weaknesses mentioned by the referees, providing more details in the text and including more figures. We expect that this revised version of our manuscript fulfills the publication standards of SciPost.

---

## Round 2 · List of Changes

Regarding the suggested major modifications from Report 1: 1. We included a discussion about a proper implementation of experimental cuts for LHC Run II. Based on the suggested reports and private communications with experimentalists, we modified Figs. 1 – 4 with more realistic cuts. A discussion about this was added in Sec. 3, as well as new comments on the phenomenological impact for ATLAS and CMS measurements. 2. The scale uncertainties were propagated to the reconstructed partonic momentum fractions, as carefully explained in Appendix A and in Sec. 4.5. We also include new plots to show the effect in the reconstruction efficiency. Regarding the propagation of the PDF/FF fitting uncertainties, previous studies cited in this work suggest that their contribution is rather small compared to the errors induced by the scale uncertainties. For this reason, we restrict our error analysis to the one discussed in Appendix A and in Sec. 4.5. 3. We give more details on the loss function in Sec. 4.4. 4. Asymptotic solutions were not computed and we do not see any straightforward method for doing that, since we do not have closed analytic expressions for the NLO cross-sections. In fact, the higher-order corrections are implemented with the FKS algorithm, which is mainly intended for numerical calculations. 5. Former Fig. 16 (Fig. 19) is intended for comparing the architectures and different parameters in the MLP. In fact, the suggestion of enlarging the network was implemented in the work to generate Fig. 15. We tried to avoid overfitting, thus we fixed the training dataset size for the different methods (80% of the total). Also, we tried to find a balance between training time and quality of the reconstruction. 6. The dataset used for training the network was then used to generate the correlation plots. At LO, given the measurable quantities V_EXP it is possible to unambiguously calculate x1, x2 and z. However, beyond LO, the presence of radiative corrections leads to events with the same pHAD and pGAMMA but different x1, x2 and z. Since the higher-order corrections are expected to be small/same order w.r.t. the LO, the effect is a spread in the correlation plots. As a consequence, even if a perfect reconstruction takes place, events outside the diagonal are expected (although less probable). An estimation of a bias in the simulation is outside the scope of the present article, since we want to test the reconstruction with a fixed simulation.

Regarding the comments from Report 2: 1. We do not fully understand the comment about model dependence. We aim to reconstruct the partonic fractions generated by our simulator (kept fixed), thus we used the same datasets for the different models. The reconstructed x1, x2 and z are model dependent, but we want to test the reconstruction quality compared with x1TRUE, x2TRUE, zTRUE from the simulator. 2. For different processes, we need to generate a different dataset and train. This is expected from the fact that the relations between (x1, x2, z) and the variables in V_EXP depend on the explicit process under consideration. The relations are not the same for pp->h+gamma and pp->gamma+gamma+jet, for instance. 3. A comment comparing the kinematics of e+e-/ep versus pp collisions is included in the main text.

---

## Round 3 · Referee Report · Anonymous (Referee 2) · 2022-7-6

Report

Thank you for taking into account my feedback. As I said in my last response, I think the paper could be significantly streamlined, but I won't insist.

Sorry, I had a typo in my last post - it should have read "It is NOT necessary to explain what a neural network is in the main body." (as I wrote in my first set of comments). I won't insist on this point, but it is odd to have such pedagogical descriptions in a paper of this type.

For the citations, I really don't think citing a website is appropriate in this case. There are plenty of review articles that would be reasonable to cite. As above, I won't insist, but please think about citing some reviews instead of websites.

---

## Round 3 · Referee Report · Anonymous (Referee 1) · 2022-8-2

Report

Dear Authors,

I thank you for adding the loss information I've requested. Stating the loss function is important because it states the desired goal of your analysis in a statistics language, and therefore, it is necessary for reproducing your results. Using a different loss function for your regression may result in estimators with different statistical characteristics (for example, mean square error vs. mean absolute error.) The physical conclusion may change if the results are getting sensitive to the characteristics.

The authors answered all the issues raised from my side, so I recommend the draft for publication.

---

## Round 3 · Author Response

Dear Editor,

We have carefully considered the comments that both referees have provided us. We have made three main modifications in line with their suggestions:

  1. Neural networks are defined already at the beginning of Section 4.4. For clarity, we have expanded their definition, incorporated the schematic representation in terms of layers and cite one of the pioneering works on the topic.

  2. We included, also in Sec. 4.4 (fourth paragraph), a comment about the cost/loss function used in our implementation. We included the reference to the documentation of the package.

  3. We replaced the figures in PNG format by EPS to increase the quality of the figures.

We hope that the modifications implemented serve to clarify the questions and address all the issues identified by the referees.

---

## Round 3 · List of Changes

Regarding the specific questions raised by the referees, we proceed to provide answers:

Referee 2:

  1. Given that this article is about phenomenology, we consider the details pertaining to the process under study to be relevant for the proper understanding of the work. Therefore, we feel that Secs. 2 and 3 belong in the body of the article rather than in an appendix, as suggested by the referee.

  2. Additional explanations about what a neural network is, are included in Sec. 4.4.

  3. We find ourselves at loss with the comment about the references used in the introduction. The purpose of the references we included, such as the workshops, was to point out the broad application of ML in particle physics and the significant interest of the community. As such, we feel that the references selected serve to our intent more than citing specific articles. Far from us to claim that these are the only works in HEP. We would be very happy to include more references if the referees were kind enough to point them to us.

  4. We included EPS graphs to improve the resolution.

  5. In Sec. 4.5 we present an estimation of the reconstruction errors due to the scale uncertainty propagation (at the partonic level). Since K-factors are usually very large for these processes, they dominate over the intrinsic PDF/FF errors (in most of the cases). Thus, we restricted our attention to the scale uncertainty propagation. On the other hand, changing the PDF/FF sets would have led to a much more complex and computationally demanding analysis, that is far beyond the scope of the current work. In fact, it would be another article altogether. Finally, regarding the discussion in App. A, we prefer to keep it as an appendix to avoid overloading the main text with rather technical (although important) discussions.

  6. The comment can be found in the second paragraph of Sec. 4, discussing the kinematics of the different collisions.

Referee 1:

  1. A comment about the implementation of the loss/cost function was included in Sec. 4.4 (fourth paragraph).

---

## Editorial Decision

published